



# Properties and Characteristics of Atmospheric Deserts over Europe

Fiona Fix-Hewitt[1], Achim Zeileis[2], Isabell Stucke[1], Reto Stauffer[3], and Georg J. Mayr[1]

[1]Department of Atmospheric and Cryospheric Sciences, Universität Innsbruck, Innsbruck, Austria
[2]Department of Statistics, Universität Innsbruck, Innsbruck, Austria
[3]Department of Statistics & Digital Science Center, Universität Innsbruck, Innsbruck, Austria

**Correspondence:** Fiona Fix-Hewitt (fiona.fix@uibk.ac.at)

**Abstract.** Atmospheric deserts are air masses that are advected from the deep, hot, dry boundary layer of arid or desert source regions. We track air masses travelling from North Africa across Europe continuously during the period between May 2022 and April 2024. The Lagrangian analysis tool LAGRANTO is used to calculate 120 h-long trajectories at every hour and a spatial resolution of 5 km in the horizontal and 10 hPa in the vertical.

Atmospheric deserts occur in up to 60 % of the time in parts of Europe. They can occur everywhere in Europe and can cover up to 55 % of the area in the domain 30 °W to 60 °E and 37 °N to 73 °N. Three typical, synoptic-scale patterns occurring during atmospheric desert events are identified from a cluster analysis. The patterns show a very zonal flow, a ridge over western Europe and a deepening trough over north-western Europe, respectively, leading to eastward or north-eastward advection of air from the source region. Typically, atmospheric deserts persist for about one day on average, slightly longer close to the source

region, but the duration and extent vary considerably with the seasons. While the 90th percentile of the duration is between one and two days for most of the domain and most of the seasons, it can be more than nine days in summer in the Mediterranean.

Atmospheric desert air frequently resides between the local boundary layer height and the troposphere, and therefore modifies the temperature profile throughout the free troposphere. The atmospheric desert air rarely enters the local boundary layer, and if it does, it happens at over high orography and in the warm season. In some regions, atmospheric deserts frequently form

a lid on top of the local boundary layer, but it only persists for less than two and a half days on average - too short to cause a heat wave.

Two main groups of air streams were found from a subjective investigation of trajectory clusters and analysis of average changes along the trajectories. One group consists of three air streams that ascend strongly. Two of the three air streams in this group become even warmer and dryer than they were in the source region through condensation and evaporation. The third, and

least frequent air stream in this group cools and dries, likely due to mixing with another air mass, especially over continental Europe in the cold season. The three air streams in the other group remain at medium altitudes, and two of them cool, either due to radiative or evaporative cooling or mixing. The third air stream in this group is the previously known 'elevated mixed layer', which almost conserves its thermodynamic properties.





## 1 Introduction

Atmospheric deserts (ADs) are air masses that are transported away from the hot and dry convective boundary layer (CBL) of semi-arid, desert, subtropical, and/or elevated source regions (name and concept first introduced in Fix et al., 2024). Diabatic processes and differential advection in the vertical gradually change their original characteristics as they are advected over hundreds to thousands of kilometres towards their target region. Similar to elevated mixed layers (EMLs), they may have an influence on the weather in the target region, e.g. due to their effect on the vertical structure of the atmosphere. Therefore,

this study aims to investigate typical characteristics of ADs over Europe as well as their evolution during the advection. ADs are a generalisation of EMLs, or their Europe-specific manifestation, the Spanish Plume. This generalisation of the concept is clearly needed, as (1) it is likely that even if the air does not remain well mixed it will still have an influence on the weather in the target region and (2) many studies about the Spanish Plume show air streams that actually originate in Northern Africa instead of the Iberian Peninsula (e.g. Sibley, 2012; de Villiers, 2020; Schultz et al., 2025b, their Table 4). Schultz et al. (2025a)

investigate the convective environment of a storm over the UK in the beginning of July 2015. They find that the steep lapse rate air in the mid and upper troposphere originates in the subtropics, rather than over the Iberian Plateau.

In the special case of an EML, the thermodynamic properties are almost conserved. It was suggested (by e.g. Carlson and Ludlam, 1968; Carlson et al., 1983; Lanicci and Warner, 1991b; Cordeira et al., 2017), that the warm, dry air from the CBL of an arid source region glides up on the cooler, moister, local CBL, forms an EML, and produces a capping inversion or so-called

lid. This lid may facilitate heat wave formation and prevent thunderstorm outbreaks (Carlson and Ludlam, 1968; Carlson et al., 1983; Farrell and Carlson, 1989; Banacos and Ekster, 2010; Cordeira et al., 2017; Dahl and Fischer, 2016; Lanicci and Warner, 1991a, b; Ribeiro and Bosart, 2018; Andrews et al., 2024). Underrunning and secondary circulations at the air mass' edge can cause violent thunderstorms to erupt along the edges (e.g. Carlson and Ludlam, 1968; Keyser and Carlson, 1984; Lanicci and Warner, 1991b; Andrews et al., 2024; Schultz et al., 2025b).

EMLs are typically identified from vertical profiles in the target region. For ADs this is often not possible, since they are usually modified during their journey from source to target. To analyse the more general case of an AD, a direct detection method is necessary. We therefore developed such a direct detection method in a previous study (Fix et al., 2024), which uses Lagrangian trajectories to trace the air mass directly from its source to its target. The case study presented in Fix et al. (2024) shows that in mid-June 2022 the AD spanned a large part of Europe for a few days. Four different pathways of the trajectories

could be identified based on changes in their characteristics between their initialisation and arrival in the target region: One behaves like an EML, almost preserving its thermodynamic properties and rising only slightly, a second one rises high, warms and dries, while the third cools, moistens and sinks after an initial ascent. The fourth one took a different path and did not necessarily reach the target region of interest. The changes experienced by the trajectories are likely dominated by phase changes of the water vapour, and precipitation. Additionally, it was shown that the AD in mid-June 2022 was accompanied by

very high temperatures in central and southern Europe, and that lightning mainly occurred close to the AD's edge (and at the cold front). However, the presence of a capping inversion facilitating heat build-up could not be confirmed.



The present study aims at generalising the results from that case study and investigating the properties of ADs over Europe during a longer time period. ADs are tracked continuously between May 2022 and April 2024, resulting in a 2-year dataset of ADs over Europe. With this dataset, we aim to answer how often ADs occur, what area they span, how long they last, what their vertical structure is, how they are modified during the advection, and how this changes seasonally and regionally.

## 2 Data and methods

The direct detection method requires the calculation of trajectories using the Lagrangian trajectory tool (LAGRANTO; Sprenger and Wernli, 2015) and meteorological input data. In this study, the ERA5 reanalysis dataset (Hersbach et al., 2020) is used, which is explained in Section 2.1. The calculation of the trajectories is explained in Section 2.2, and the identification of the air mass in Section 2.3. How synoptic charts and trajectories can be clustered is explained in Section 2.4. In Section 2.5 we define when an AD is considered to form a lid.

The focus period of this study is 01 May 2022, 00 UTC to 30 April 2024, 23 UTC. We are aware that this study only covers a 2-year period. However, the computational and storage demand of the direct detection method are high (about 25TB for this study). Nevertheless, ADs have never been studied before (apart from the case study in Fix et al. (2024)) and the 2-year dataset is already able to give insight in many of their characteristics, especially because they do occur rather frequently.

### 2.1 Reanalysis dataset, ERA5

The latest global reanalysis from the European Centre for Medium-Range Weather Forecasts (ECMWF), ERA5 (Hersbach et al., 2020), is used as a spatio-temporally complete set of atmospheric data. It is based on the Integrated Forecasting System (IFS) Cy41r2, has a horizontal resolution of $0.25°$, and data are available hourly on 137 vertical model levels up to 1 Pa (Hersbach et al., 2020; European Centre for Medium-Range Weather Forecasts, 2016). The vertical resolution is therefore about 20 m at the surface and 300 m at 500 hPa. ERA5 single-level, pressure-level and model-level data on the lowest 74 model levels (surface to about 120 hPa) are utilised. In this study, a domain covering Northern Africa and Europe is chosen, specifically $30°$ W to $60°$ E and $15°$ to $73°$ N.

### 2.2 Trajectory calculation

The Lagrangian analysis tool (LAGRANTO) version 2.0 is used to calculate forward trajectories in this study. It has been developed since the late nineties (Sprenger and Wernli, 2015) and is a mature and widely used tool in atmospheric science, in various contexts (e.g. Stohl et al., 2001; van der Does et al., 2018; Keune et al., 2022; Oertel et al., 2023). The 3D wind field of the input dataset is used to calculate trajectories iteratively.

In this work, we are interested in ADs over Europe. Naturally, North Africa is the source region of interest. For the case study in Fix et al. (2024) it was shown that the proportion of trajectories originating in Iberia is very small, and literature also suggests, that often the air involved in Spanish Plumes is actually of subtropical origin (e.g. Schultz et al., 2025a, b), therefore we neglect Iberia as an additional source region. Trajectories are initiated along a 'curtain' marking the northern boundary




of the source region (North Africa, Fig. 1d), as all trajectories that start within the source region and reach Europe must pass through this curtain. Trajectories are started at a very high resolution of approximately 5 km in the horizontal, and 10 hPa in the vertical between 1100 and 400 hPa. The approach differs slightly from the one described in Fix et al. (2024) and requires much fewer trajectories without affecting the results as tested for that case study. The results in Fix et al. (2024) are qualitatively the same whether trajectories are only started from the BL during daytime hours or from a smoothed BL at all times, but for this study continuos initiation was desired.

Per definition AD air should originate in the source region's boundary layer (BL). The boundary layer height (BLH) in desert regions can grow up to several kilometres during the day and has a very strong diurnal cycle (e.g. Garcia-Carreras et al., 2015). It can be assumed that the night time residual layer has similar properties as the daytime CBL and can also contribute to ADs. In this study, we require a continuous initiation (hourly) of trajectories. There is no reliable measure for the top of the residual layer in ERA5, so we use an interpolated $BLH_i$, where $BLH_i$ is interpolated between the daily maxima of the BLH.

We initiate trajectories from below $BLH_i$ at the curtain, and calculate them forward in time only if they have a northward wind component (and therefore have a chance of arriving in Europe). The trajectories are calculated 120 h into the future, which is a reasonable choice, making sure that the trajectories can reach everywhere in the domain within that time, but keeping them short enough so that they have not fully lost their former CBL properties. This choice is supported by the results of the following tests: The average age of the trajectories when they reach the northern regions of the domain is less than 120 h and the comparison with experiments using 168 or 240 h long trajectories did not show much difference in the air mass extent.

Trajectories are initiated every hour between 01 May 2022, 00 UTC and 30 April 2024, 23 UTC. This results in approximately 108 million trajectories, 61 million of which pass north of 37° N and therefore contribute to the analysis. Height above mean sea level (h. a.m.s.l), potential temperature ($\theta$), specific humidity ($q$), and equivalent potential temperature ($\theta_E$) are traced along the trajectories. For time steps where $q$ is negative in ERA5 (due to numerical inaccuracies), $\theta_E$ is set to $\theta$, however, this only affects less than 1 % of all trajectories.

## 2.3 Detection of the atmospheric desert air mass

In order to identify the AD air mass, the trajectories are aggregated to grid boxes of 0.25° x 0.25° x 500 m, matching ERA5 grid cells in the horizontal. 500 m in the vertical are chosen so that a decent vertical resolution is achieved, but high enough numbers of trajectories can be aggregated into each box. An AD-cell is then defined as a grid box that contains at least one trajectory. This results in a dataset that designates each point in space and time as AD- or nonAD-cell. Additionally, the average properties of all trajectories within that cell are known. Using only one trajectory as threshold to identify a cell as AD-cell may seem like a weak definition, but as argued in Fix et al. (2024) it is a useful one and was shown not to substantially misidentify the AD-cells. A column is an AD-column if there is at least one AD-cell in this vertical column. They therefore have a horizontal extent of 0.25° x 0.25°, as the ERA5 grid cells.

The AD streak length is defined for each ERA5-cell as the duration of continuous presence of an AD-column (AD air anywhere in the column). Since the AD data can be noisy, especially around its edges, short gaps of 1 h were filled, so that the average persistence is not underestimated if long periods are interrupted briefly.



Maxima of AD-events are defined as the time with maximal horizontal extent. They are identified as the local maxima in the smoothed time series of the percentage of AD-columns north of 37 °N. The time series is smoothed using a 4th-order Butterworth low-pass filter with a cut-off frequency of $1/2\mathrm{d}$. This removes noise associated with variations occurring at frequencies

higher than every 2 days. Then, local maxima with distances of at least $1\,\mathrm{d}$ and prominence of at least 0.05 are detected. Similarly, the onset of an AD-event is determined as the local minima in that time series.

## 2.4 Clustering

In order to analyse typical synoptic patterns associated with ADs, geopotential height maps are clustered. This is done for the 66 AD-events with at least 2 days between their onset and their maximum. We focus on geopotential height at $500\,\mathrm{hPa}$

($geoh_{500}$) at $24\,\mathrm{h}$ prior to the AD maximum. This allows to identify and categorise large-scale atmospheric circulation patterns linked to ADs. First, the $geoh_{500}$ maps are normalised to remove the influence of absolute values and focus on the relative spatial patterns. The data is then reshaped to a spatial dimension (space = lon x lat). To facilitate computation, the data is smoothed using a principal component analysis retaining 95 % of the variance. Then they are clustered using $k$-means clustering (MacQueen, 1967; Cos et al., 2025). The suitable number of clusters is usually chosen under the consideration of the total sum

of squares. The aim is that it does not decrease much further with additional clusters. However, also the number of members per clusters, as well as their interpretability should be kept in mind. For this study we find three clusters to be a reasonable choice.

Additionally, 'typical' trajectory paths and developments are of interest. Analysing the route and the development en-route of the trajectories can give insight into the processes modifying the AD air during the advection. To reduce the dimensionality of the problem, all trajectories initiated on one day are clustered using $k$-means clustering. Trajectories are clustered by the

following 5 variables: their differences in longitude, latitude, height above mean sea level, potential temperature, and specific humidity between their initialisation and their end at $120\,\mathrm{h}$ (note that trajectories that leave the domain are not considered). This is a similar approach as in Fix et al. (2024) but with reduced number of variables. This choice was made, so that the variables used do not over-emphasise phase change driven processes in the clustering. The variables are standardised to ensure

equal weight and a number of 4 clusters is chosen for the clustering (as in Fix et al., 2024). This clustering is performed for every starting day between $24\,\mathrm{h}$ after the onset and before the maximum (290 days in total) of all 66 AD-events, as well as the 5th day before each of the maxima (as this is the trajectory length). From the daily cluster averages, 'typical' ones that occur similarly on several days are identified subjectively and one representative is chosen for each 'typical' cluster. This analysis provides a qualitative understanding of the most common trajectory patterns. By focusing on representative, 'typical' cluster

averages, we aim to highlight the dominant mechanisms influencing the AD air during advection, rather than the magnitudes of individual changes.

## 2.5 Lid

ADs may form a lid on top of the local BL, which can have consequences for the local weather. We consider the AD to have formed a lid if the centre of the lowest AD-cell lies within $\pm500$ m of the BLH. This may seem like a very loose criterion,





but inspection of the vertical temperature profiles and the BLH in ERA5 have shown that the ERA5 BLH variable sometimes differs from where the potential temperature profile would suggest it by several hundred metres. It is known that ERA5 BLH is not without problems (e.g. Madonna et al., 2021; Wei et al., 2025). Note that the lid defined like this does not necessarily indicate the presence of a capping inversion, only the possibility of one caused by an AD sitting directly on top of the local BLH.

The lid-criterion is less meaningful during nighttime, however, as the BLH can be very low, and the residual layer cannot be properly taken into account. We therefore focus the lid analysis on the daytime hours only (6 hours, 10 through 15 UTC). The streak length of the lid in a ERA5 grid cell is defined as the number of consecutive days which had a lid during at least 3 (half) of the daytime hours.

## 3   Results

In this section the properties of ADs are described. Their occurrence frequency and spatial extent are investigated (Sect. 3.1), as well as the synoptic patterns leading to AD-events (Sect. 3.2). Especially with respect to possible impacts, the persistence of AD-events (Sect. 3.3), their vertical structure, their ability to form a capping inversion, and the duration of the latter are of great interest (Sect. 3.4). Furthermore, the development of the AD air during the advection and the related physical processes are analysed (Sect. 3.5).

### 3.1   Occurrence frequency and spatial extent of ADs over Europe

The first question to answer is how often ADs occur. The presence of an AD in a respective column is a binary time series, hence the probability can be determined as its temporal mean. Due to Earth's curvature, ERA5 grid cells at high latitudes are smaller than those at lower latitudes. Therefore, the probability seen in Fig. 1a is weighted by grid cell area, which gives more weight to the cells further north and facilitates the interpretation of the occurrence pattern. The region with elevated AD prob-

ability extends from the Iberian peninsula north-eastward, as far as northern Scandinavia. The area-weighted AD probability reaches more than $0.7 \text{x} 10^{-3} \text{ km}^{-2} \text{h}^{-1}$ in the southern Mediterranean, $0.4 - 0.6 \text{ x} 10^{-3} \text{ km}^{-2} \text{h}^{-1}$ in central Europe, and still up to $0.3 \text{x} 10^{-3} \text{ km}^{-2} \text{h}^{-1}$ in parts of Scandinavia. Over the British Isles, the weighted probability is less than $0.2 \text{x} 10^{-3} \text{ km}^{-2} \text{h}^{-1}$. The weighted numbers might not be intuitively interpretable, but highlight the occurrence pattern well. Non-weighted probabilities for an hour to be an AD-hour exceed 60% in the Mediterranean, and are up to 20% as far north as southern Scandinavia

(not shown). Hence, ADs are frequent and can occur almost everywhere in Europe.

ADs cover between 0 and 55 % of the area in the domain, the median of the 2 years being about 15 % (see Fig. 1c, allyear). This behaviour varies with the seasons. In boreal winter (DJF) and boreal summer (JJA) the maximum and median extent is smaller. In DJF, the maximum probability is recorded in southeastern Europe (Fig. 1f). The maximum AD probability in JJA is concentrated in the western Mediterranean (Fig. 1e). Very low probabilities are recorded in eastern Europe in JJA,

and events spanning less than 10 % of the area are most common. Hence, while ADs are very frequent in JJA in the western Mediterranean, they tend to be spatially confined to the southern latitudes of the domain. The highest probabilities and largest





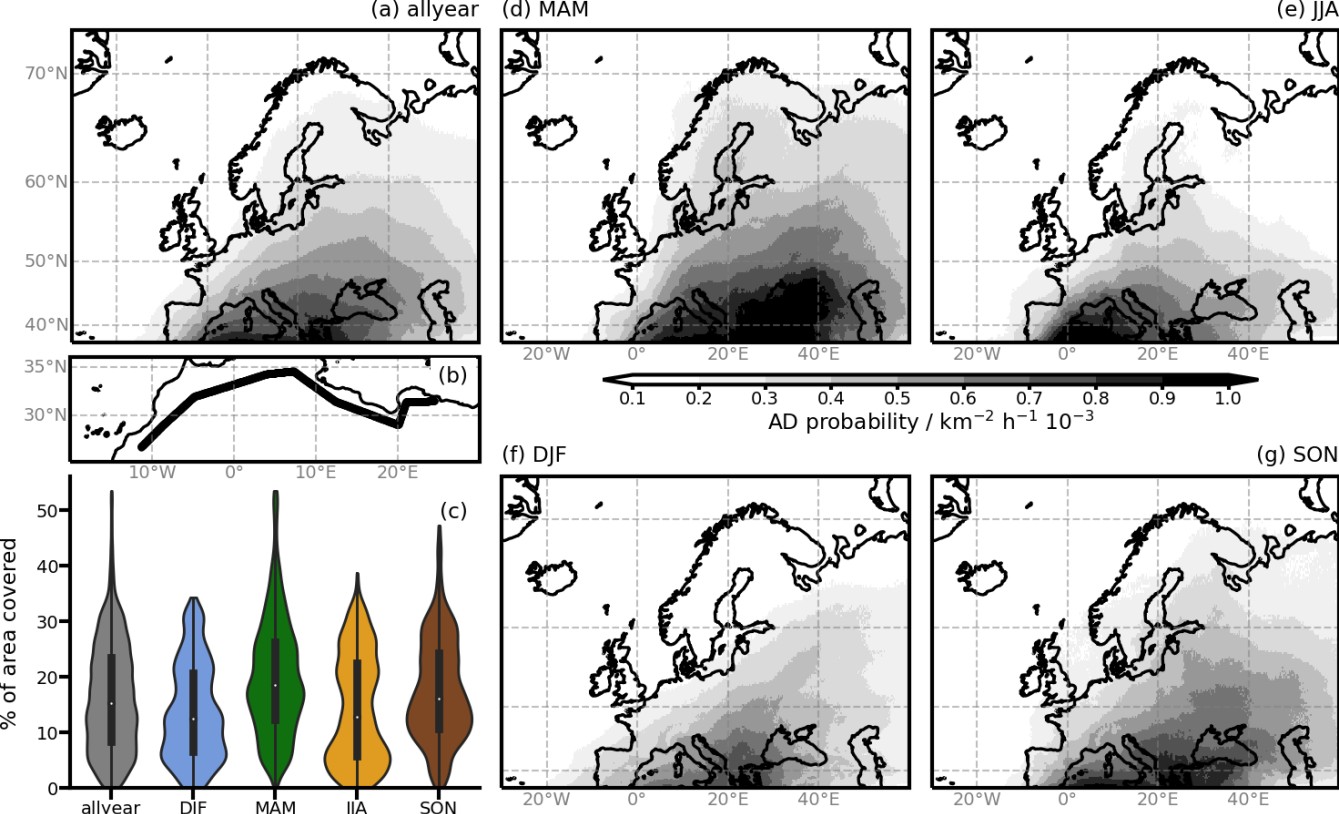

**Figure 1.** (a) Probability of an AD being present in the respective cell, weighted by the cell's area, for the entire 2-year period (allyear). (b) Locations of the starting points of the trajectories along the 'curtain'. (c) Violin plot of the percentage of the area in the domain covered by an AD, per season. The violins are normed to have the same area. (d)-(g) Same as (a), but for MAM, JJA, DJF, and SON, respectively.

events are recorded in boreal spring (MAM), with highest probabilities especially in the eastern Mediterranean, over Turkey and the Black Sea (Figs. 1c,d). Both in the probability map (Fig. 1d) as well as the violin plot (Fig. 1c) it becomes clear, that MAM is also the season with the largest AD-events, spanning up to 55 % of the area, with noticeable probabilities recorded

over all of Scandinavia. The median size of ADs in MAM is about 20 %, which is also the most frequent size in this season, whereas very small events are less likely. In boreal autumn (SON), the median size is slightly lower, but due to one peak in smaller (between 10 and 15 %) and one in larger (around 30 %) events (Fig. 1c). Also in SON very small events are less likely. Probabilities are elevated across the entire Mediterranean in SON and low but noticeable probabilities are recorded over the British Isles and Scandinavia (Fig. 1g).

A more detailed insight in the AD extent can be achieved when looking at the temporal evolution of the extent of the ADs (see Fig. 2), which shows a time series of the AD extent. Long, vertical stripes depict ADs that reach far north, while darker colours depict ADs that cover a wide longitudinal band. While this figure does not allow a detailed analysis of the AD extent, it does show that it is rare that no AD air is present in the domain north of 37° N. Many times during these 2 years, ADs do reach





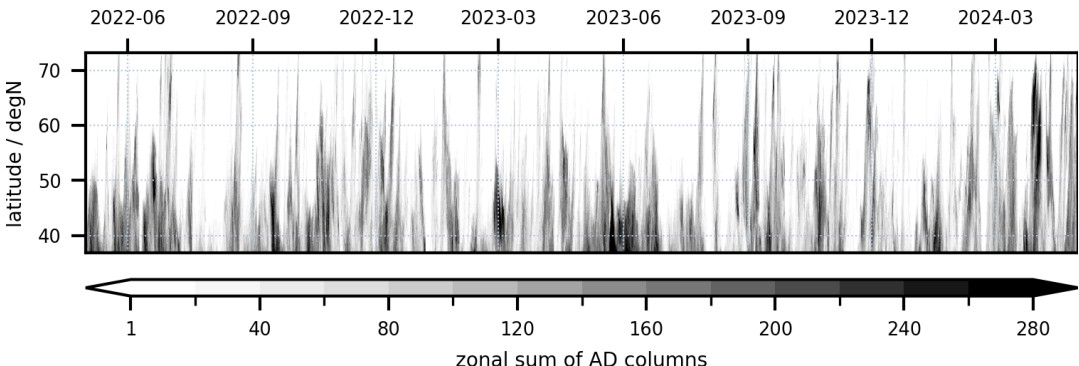

**Figure 2.** Time series of AD-presence. Time is shown on the x-axis, the y-axis shows latitude, shading shows the number of columns at that latitude, which have an AD present. There are 361 columns at each latitude in the domain.

as far north as Scandinavia, and there are very different flavours of events, i.e. different extents towards the north, different
widths (intensity of the colour in Fig. 2), and different durations. This also becomes apparent, when additionally looking at
the number of trajectories in the domain (not shown here) and not only the number of identified AD-columns. A large spatial
extent can occur both with a high and low trajectory density. The same is valid for ADs with a smaller spatial extent.

## 3.2   Synoptic conditions

In order to better understand the synoptic conditions that lead to ADs, geopotential height maps at 500 hPa are used. All 66
identified AD-events (Sect. 2.3) are clustered by $geoh_{500}$ at 24 h before the maximum aerial extent (Sect. 2.4). This time is
chosen as representative for the synoptic situation during a fully developed AD-event, but early enough so that it does not
yet represent the pattern present during the AD decay. The clustering results in three synoptic-scale patterns that are related
to AD-events. Cluster averages for 24 h after the onset of an AD period (= local minimum in smoothed time series of AD
coverage) are also shown as representative for the pattern during the early stages of the event, when advection towards Europe
begins.

   The first cluster of geopotential heights (Fig. 3, left) shows a very consistent, zonal pattern in the geopotential height. This is
also in line with the visual inspection of the typical trajectory cluster average paths (Sect. 2.4, not shown here), some of which
travel strictly east or clockwise through the Mediterranean, travelling north-easterly in the western Mediterranean Sea, and
then south-easterly in the eastern Mediterranean Sea. The second cluster (Fig. 3, middle) shows a ridge over western Europe
that intensifies strongly between 24 h after onset (top) and 24 h before the maximum (bottom). This pattern likely leads to paths
of trajectories that travel northwards across western Europe and then turn east over northern Europe following the shape of the
ridge, as was seen in Fix et al. (2024). The third cluster (Fig. 3, right) shows a deepening trough over north-western Europe,
which leads to the trajectories being transported north-east in front of it, sometimes with a cyclonic rotation. Again, this fits
the paths seen from some of the trajectory cluster averages as well. These patterns and the associated trajectory paths explain




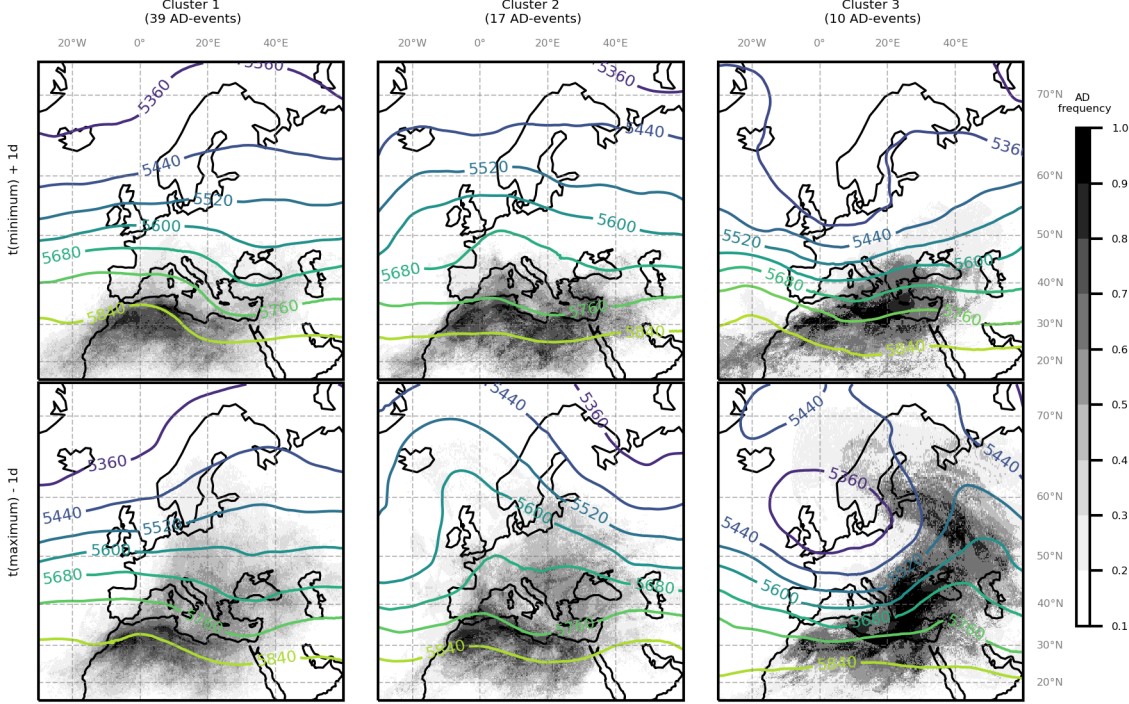

**Figure 3.** Cluster averages of $geoh_{500}$ in m (contours) and AD frequency (h$^{-1}$, shading). The top row shows the cluster average at 24 h after the onset, and the lower row at 24 h before the maximum aerial extent. The mean AD presence (= AD frequency) per cluster is shown as shading for context. Clusters are based on $geoh_{500}$ only at 24 h before the maximum (contours in bottom row). The number of members in each cluster is given in the column title.

the pattern of highest frequencies seen in Fig. 1a well, as all patterns leading to AD-events cause an easterly or north-easterly advection.

### 3.3  Persistence of ADs over Europe

Another important feature of ADs, especially with regard to their consequences, is their persistence. ADs over the Mediterranean prevail for about 1 to 2 days on average, while in most of the rest of the domain the average streak length is less than 225  a day. Maximum streak durations of more than 4 weeks were reached in the Mediterranean, while in central Europe the maximum streak length is around a week. However, the maximum streak length may only represent a single event for large parts of the domain, so the 90th percentile may be a more robust measure to look at the longest streaks. The 90th percentile of AD streak lengths is between 2 and 3 days for most of the domain, and more, but still less than a week, in the Mediterranean.





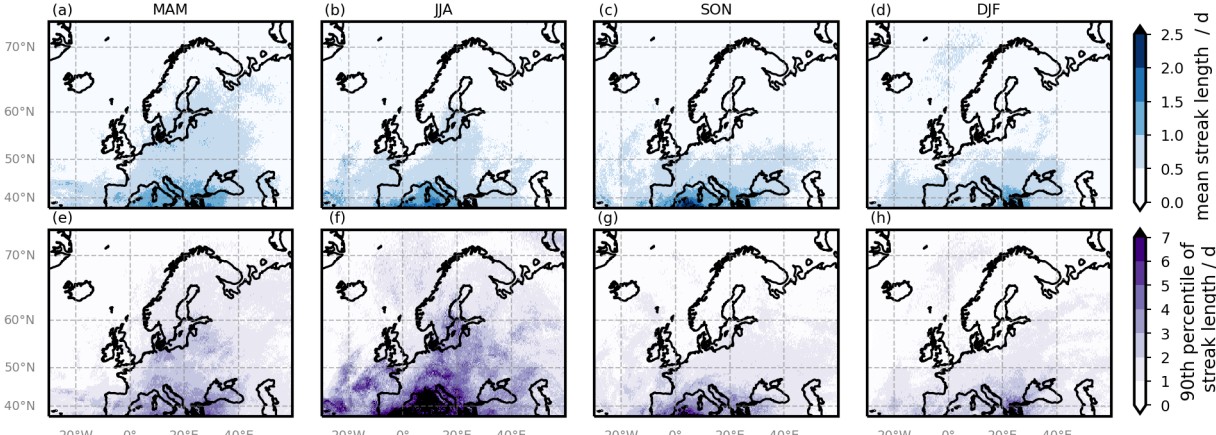

**Figure 4.** Mean (top) and 90th percentile (bottom) AD streak length in days, per season (columns).

A more detailed view is given in Fig. 4, where the mean (top) and 90th percentile (bottom) streak lengths are shown per
season. The mean streak lengths in each season are in a similar range and the pattern resembles the occurrence probability
in the respective seasons (Fig. 4, top row, compare Fig. 1d-g). The longest average durations are recorded in the western
Mediterranean in SON. The 90th percentile of streak length varies strongly across the seasons (Fig. 4, bottom row). In most
seasons, the 90th percentile of the streak length is well below a week, closer to a week in the Mediterranean in SON. During
JJA, however, 90th percentile streak lengths of more than nine days are reached in the Mediterranean, maxima are even up
to four weeks. This indicates, that ADs are not only frequent in the Mediterranean in JJA, but can also be very persistent.
Contrary, MAM showed high probabilities and large extents, but the events seem to be less persistent.

### 3.4 Vertical structure of ADs over Europe

Not only the horizontal but also also the vertical extent and the vertical structure of ADs are of interest, especially with regard to
the influence of ADs on local weather. On average, the ADs are about 2-8 cells, i.e. 1-4 km thick (Fig. 5c). Also, the ADs seem
to be coherent in the vertical, in the sense, that mostly the entire column between the lowest and highest cells are filled with
AD-cells, interruptions in the vertical are few and small (not shown). On average, the lowest AD-cell is between approximately
2 and 6 km, and the highest between approximately 4 and 8 km (Fig. 5a,b). Both in the distribution of the lowest and the highest
AD-cells, it becomes apparent that ADs are on average tilted upwards towards the north-east. This is sensible, considering they
are advected north or north-eastward, where they ride up on colder, moister, local air masses. The largest average distance
between the highest and lowest AD-cells is found over central Europe (Fig. 5c). The overall pattern of the lowest and highest
AD-cells looks similar in all seasons (more detail in the Appendix, Fig. A1).

EMLs have been argued to cause heat build-up in the BL below, and prevent thunderstorms in their centres due to a strong
capping inversion or lid that separates the BL below from the warmer, dryer EML air above (e.g. Carlson and Ludlam, 1968;
Lanicci and Warner, 1991a, b; Cordeira et al., 2017). We therefore investigate how often ADs form a lid, how long it persists,



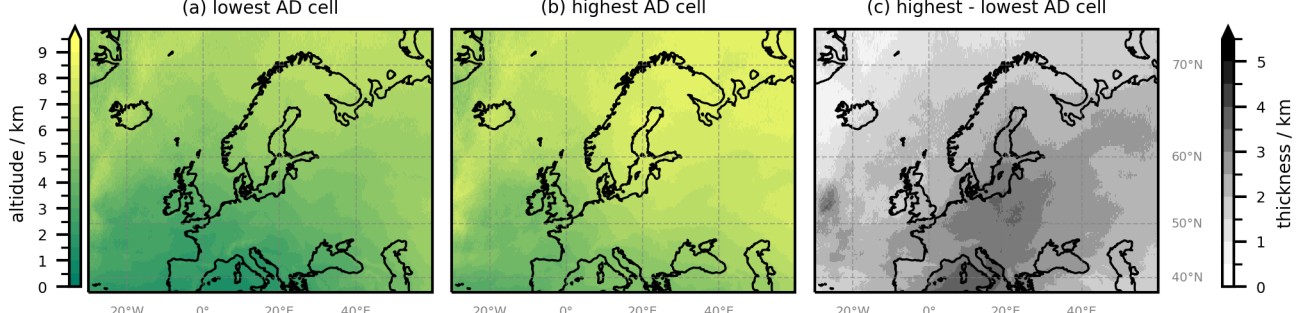

**Figure 5.** Average altitudes of lowest (a) and highest (b) AD-cells, and their distance (c). This distance describes the maximal thickness of the AD, but it does not depict the actual number of layers present.

and whether and how often AD air enters the local BL. This analysis shows that a lid is present in up to 45 % of the daytime AD cell-hours in the western Mediterranean (Fig. 6a). South of 55°N and west of 40°E lid probabilities are still elevated, with about 20-40 % of all AD cell-hours having a lid. Beyond this, the probability of an AD causing a lid is much reduced, which is in line with the AD base height being higher, the further north and east it gets (see Fig. 5a and Section 3.1). Hence, the pattern is not the same as the pattern of the AD probabilities, and the probability of an AD forming a lid therefore differs between

regions (Figs. 6 and 1a).

The impact of an AD on the local weather may strongly depend on the persistence of such a lid. We therefore analyse the average and 90th percentile of the streak length of the lids (Fig. 6b,c). It becomes very clear that in most of the domain the average streak length is between 1 and 2 days. Only in the Mediterranean over the sea the average streak length is between 2 and 3 days, with few cells exceeding 3 days. Another area with long average lid persistence, but similar 90th percentile streak

length, is found between Iceland and northern Norway. This is an area with low AD frequency, hence few events with durations of around 2 days might dominate the average. Also the 90th percentile of streak lengths is around a few days in most of the domain. Only in the Mediterranean, the 90th percentile exceeds a week. The maximum duration of a lid there is even up to 2 weeks, in some cells even longer (not shown here). It becomes very obvious that the probability as well as mean and 90th percentile of the duration of a lid is elevated over the sea (Mediterranean, Black Sea, and the Atlantic off the Iberian Coast).

This likely has to do with the different properties of a marine BL compared to a continental one, as it is often cooler and moister. Highest probabilities of a lid are recorded in SON and DJF, with probabilities up to more than 55 % in the Mediterranean and west of Iberia (more detail in the Appendix, Fig. A2).

Additional to whether the AD resides just on top of the local BL and forms a lid, it is of interest, whether AD air enters the local BL at all. An AD-cell is identified as within the BL, when its centre is below $BLH - 500\,\mathrm{m}$ (to match the definition of the

lid, also daytime hours only). The probability of an AD-cell within the BL given that an AD is present is shown in Fig. 7. AD air penetrates the local BL in less than 20 % of the time in most of the domain. This is expected, as AD air is expected to have higher potential temperature than the local BL, simply due to its origin. However, over the Iberian highlands this probability is greatly increased, to more than 60 % of the daytime cell-hours. Similar behaviour is seen in the Pyrenees, Alps, Apennines,





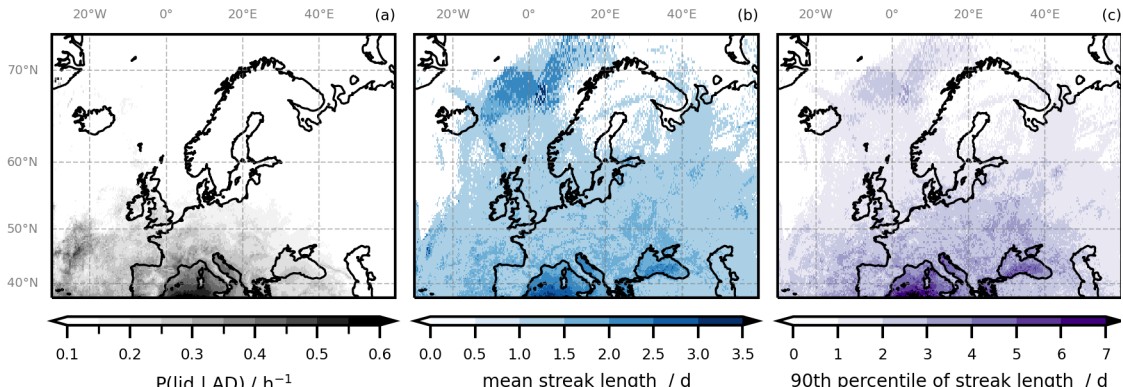

**Figure 6.** Lid properties. (a) Probability of a lid being present, given an AD is present, P(lid | AD). (b) Mean streak length of the lid in days. (c) 90th percentile of streak length of the lid in days.

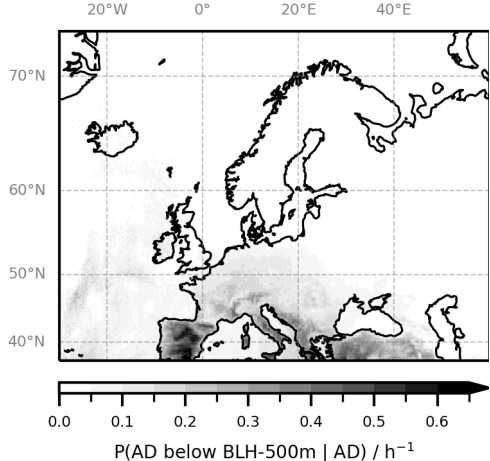

**Figure 7.** Probability of the the lowest AD-cell being within the BL, P(AD below BLH | AD).

and Dinaric Alps - hence, in mountainous regions. This is due to the fact that in these regions, simply due to orography, the
BLH is already higher above sea level than in other regions. With enough solar heating its properties might become similar to
those of the arriving AD air, which facilitates the entering of the latter. This is supported by the seasonal analysis (Fig. A3, in
the appendix).

Hence, air of desert origin can populate the free troposphere between the local BLH and the tropopause in the target region
frequently (Figs. 5 and 1). It rarely penetrates the local BLH (Fig. 7). Therefore, the temperature profile in the free troposphere
in the target region is often modified by the presence of an AD. Sometimes the AD also forms a lid on top of the local BL,
however, the persistence of these lids is typically short, so that they are not likely to cause considerable heat build-up and hence
heat waves.





## 3.5 Average changes during advection

The changes an AD experiences during its advection from source to target are also of interest in order to understand ADs.

Hence, in every AD-cell, the difference in potential temperature, $\theta$, specific humidity, $q$, and equivalent potential temperature, $\theta_{\mathrm{E}}$, between the current value and the trajectories' starting point, as well as the trajectories' age are averaged over all trajectories in the respective AD-cell. Note that every AD-cell ($0.25°$x$0.25°$x$500\,$m) can contain any number of trajectories at a given time. The different trajectories in the cell may have come there along different paths, and taken different times to get there, but their combined properties decide the properties of the AD at this location, which is why the cell averaged value is a reasonable

choice. The distribution of these average changes is depicted in Fig. 8 for all cells north of $37°\,$N and all time steps.

The average changes per AD-cell are interpreted in synergy with the average development of the 'typical' trajectory clusters (Sect. 2.4, same as in the evaluation of their paths in Sect. 3.1). Those 'typical' trajectory clusters are shown in Fig. 9 schematically. The figure is meant schematically, hence no scales are shown, to account for the fact that the different features may occur at different times or magnitudes. Each panel corresponds to one cluster, showing the temporal evolution of height above mean

sea level (h.a.m.s.l, solid), $\theta$ (dotted), and $q$ (dash-dotted). Time series of the difference in h.a.m.s.l, $\theta$, and $q$ can give insight into what changes the trajectories experienced during the advection.

It becomes apparent that most AD-cells are found between 2.5 and 3 km, and another favoured altitude is between 7 and 8 km (see Fig. 8a). Accordingly, there are 2 larger groups of typical trajectories: for one, trajectories remain at medium altitudes (Fig. 9, right column), and for the other they rise significantly (Fig. 9, left column). Trajectories of all ages can be found at

all altitudes (Fig. 8b). It is logical that there are more older trajectories (higher values in the right of Fig. 8b), since they had more time to arrive in the target region and remain there unless they leave the domain again during their 120 h. Figs. 8c and d show a strong relation between the change in the potential temperature, $\theta$, as well as the specific humidity, $q$, and the altitude. Trajectories that reach higher altitudes have warmed on average, up to 20 K, and lost up to $5\,$g$\,$kg$^{-1}$ of specific humidity. This behaviour can be explained when looking at the typical trajectory development in Fig. 9: Clusters a and b experience a

strong ascent (more abrupt in the case of a), together with an increase in $\theta$ and a decrease in $q$, which indicates that the ascent caused condensation which precipitates out (Fig. 9a,b, blue, purple). At least one of these clusters appears almost in all cases, independent of the geospatial path. The ascent seems to happen at a colder air mass, or sometimes at the Atlas mountain range. This air stream resembles a warm conveyor belt, but unlike a warm conveyor belt does not have its origin in warm and moist regions over subtropical oceans, but in the warm and desert source region.

On the other hand, trajectories that end up at lower altitudes, cooled considerably, and gained up to about $5\,$g$\,$kg$^{-1}$ in specific humidity (Fig. 8c,d). This fits well with cluster d (Fig. 9d, red), which also appears in many cases and descends (sometimes after an initial ascent), while cooling and moistening. This indicates evaporative cooling as the reason for the changes. This can be re-evaporation of the precipitation falling through. As this cluster is often (not exclusively) observed over the Mediterranean Sea, it could also be evaporation from the sea or mixing with the cooler, moister BL there.

There is a region between $2\,$km and $6\,$km where $\theta$ and $q$ remain almost unchanged (Fig. 8c,d). This may be an effect of averaging trajectories with different changes. It can also be that there is in fact a portion of trajectories that retain their original





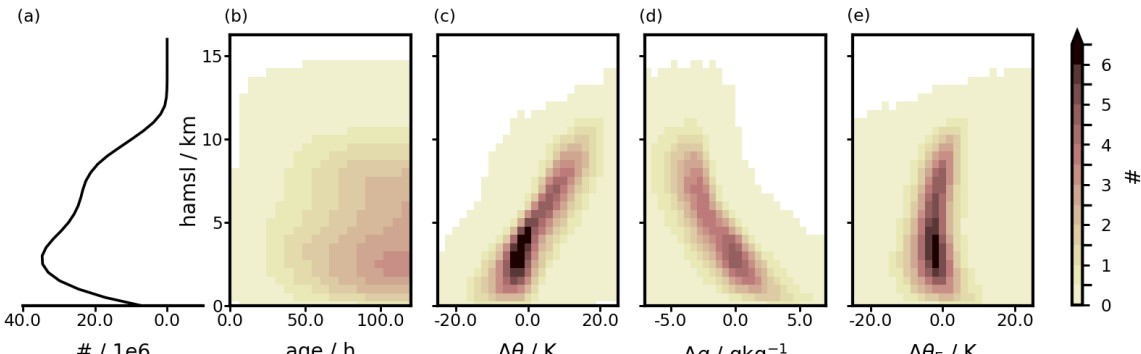

**Figure 8.** 2D-histograms of cell-averaged changes in trajectories. For each AD-cell ($0.25° \times 0.25° \times 500$ m) all trajectories within are averaged. The histograms depict the distribution of cells in total (a), the average age (b), average differences to the starting location of potential temperature ($\Delta\theta$, c), specific humidity ($\Delta q$,d), and equivalent potential temperature ($\Delta\theta_{\mathrm{E}}$,e), respectively, across all cells north of $37°$ N and all time steps.

properties, remain well-mixed and therefore form an EML. The latter is supported by the existence of cluster e (Fig. 9e, cyan), which almost conserves its $\theta$ and $q$. This cluster behaves like an EML, rarely reaches northern or eastern Europe, mainly appears in the warm season, but and is generally observed infrequently.

The last two typical clusters are less frequent and therefore not so visible in the histograms (Fig. 8). Cluster c (Fig. 9c, yellow) occurs mostly in the cold season and mainly in the eastern part of the domain. It rises (although less than a and b), cools, and dries simultaneously, which indicates mixing with a cooler, dryer, continental European air mass. The cooling is not always strongly pronounced. Cluster f does not rise high and its moisture remains constant, however, this cluster cools by a few degrees during the advection (Fig. 9f, green). The amount of cooling (a few K during the 5 days) seen here without a change

in moisture points to radiative cooling as cause.

Looking at the temporal average of the spatial distribution of the changes at several altitudes leads to a similar conclusion (not shown here). At higher altitudes the trajectories in the AD-cells have warmed and dried on average, while at lower altitudes they have cooled and moistened. There is no noticeable spatial pattern in the horizontal. Logically, the average age of the trajectories in a cell increases towards the north and east, simply due to distance to the source. On average, trajectories in cells

in northern Europe at 9 km are younger than those at 4.5 km, due to higher wind speeds at higher altitudes.

In summary it can be said that there are two main groups of air streams: One ascends and the other one remains at a similar altitude, most AD-cells are found around 8 km and 2.5-3 km altitude. The air streams that rise can become even warmer and drier than they already were in the source region, due to condensation and fall-out of precipitation. In the cold season one of the clusters in the rising group cools and dries, likely due to mixing with another air mass. The air streams that do not rise

either remain unchanged and behave like an EML, or they cool, either through radiative or evaporative cooling, or mixing.



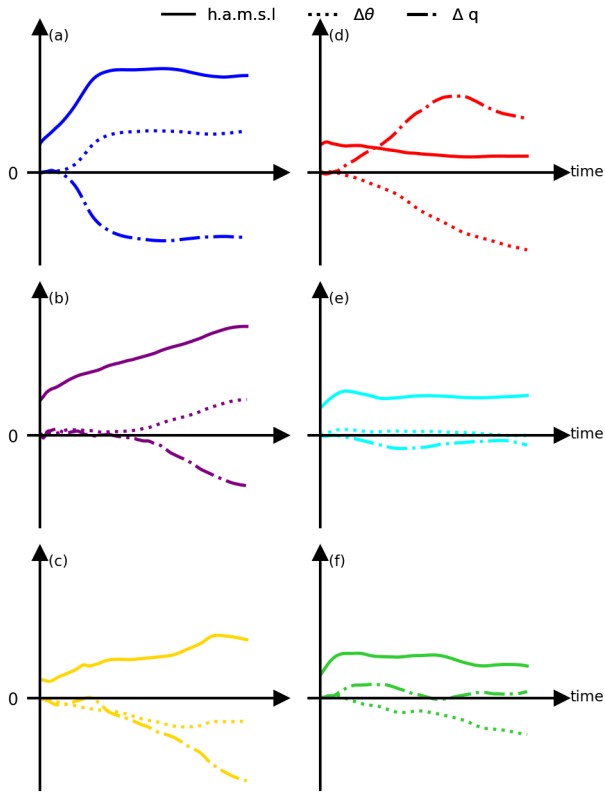

**Figure 9.** Schematic of subjectively identified 'typical' trajectory clusters a-f. x-axis shows time. No scales are given, as this figure is meant to be schematic and should account for different features to occur at different times or magnitudes. Solid, dotted, and dash-dotted lines depict differences in h.a.m.s.l, $\theta$, and $q$ since initialisation, respectively.

## 4 Conclusions, discussion and outlook

In this study, we investigated the properties of atmospheric deserts (ADs) over Europe during the period May 2022 through April 2024. ADs are air masses, that originate in the CBL of arid, desert source regions and are advected towards a typically moister, cooler target region. We employ a direct detection method tracing the air masses directly from their source region

in northern Africa to their target region in Europe. This is done using LAGRANTO, based on ERA5 input data, to calculate trajectories at very high spatio-temporal resolution.

It was shown that ADs are actually not a rare phenomenon at all. Some regions experience AD presence in up to 60 % of the cell-hours during the observed period. They can affect entire Europe and can span up to 55 % of the domain's area, the median being 15 %. Their average duration is a few days. They are more likely to occur close to the source region, where they also

tend to last longer. AD behaviour varies across seasons. Boreal spring is the season with the ADs with the largest extent and highest probabilities in parts of northern and eastern Europe. In summer, the ADs tend to have smaller extents, often confined





to the Mediterranean, but the persistence is largest. During autumn, larger events extending further east are more likely again. Boreal winter is the season with the lowest probabilities of AD in the entire domain, durations are shortest, and small extents are favoured.

A cluster analysis of the geopotential height pattern at 500 hPa 24 h prior to 66 AD maxima yielded three different synoptic situations that foster large AD-events: A very zonal structure transporting desert air eastwards, an amplified ridge over western Europe leading to northward advection over Iberia and the western Mediterranean, and a trough over western Europe, leading to north-eastward advection of desert air across the Eastern Mediterranean and over north-eastern Europe. This highlights that ADs are a frequent and variable phenomenon occurring under different flow patterns, which can possibly influence local
weather often, and in a variety of ways.

One such mechanism can be through the modification of the temperature profile in the target region. ADs often reside above the local BL and up to the troposphere, so that they modify the profile of the entire free troposphere in the target region. A particularly strong influence can be expected when the AD forms a lid directly on top of the local BL, and therefore possibly cause a capping inversion. We analysed how often this is the case during daytime. North of 55°N this is almost not happening,
further south the probability of a lid is 20-40 % given an AD is present. However, persistence of such a lid is less than 2.5 d in most of the domain, and only over the Mediterranean Sea the 90th percentile of streak lengths reaches up to a week. Hence, a considerable proportion of all AD-events can produce a lid, which can have important implications for the local weather, as it can inhibit thunderstorm eruption and foster heat accumulation underneath. We assume the latter to be unlikely, however, since the persistence of the lids are typically so short. AD air only enters the local BL rarely, and if it does it occurs over high
altitude mountainous CBLs.

Two main groups of 'typical' air streams were identified: In one group, the trajectories ascend on average, while in the other they remain at medium altitudes. Different 'typical' air streams were subjectively identified, and used in synergy with the average changes in the thermodynamic variables per AD-cell, to explain changes the AD air experiences during advection. Two of the air streams in the rising group become even warmer and dryer during the advection than they already were in the
source region. This is likely due to condensation and subsequent precipitation. The third air stream that rises, likely mixes with a cooler, dryer air mass, especially in the cold season over continental Europe. Two air streams that do not rise, cool either through radiation, evaporation, or mixing. The last air stream almost conserves its thermodynamic properties. While this analysis is more detailed and different air streams emerge in different cases, this agrees overall with the findings from Fix et al. (2024).

ADs therefore include the two well-studied special cases of EMLs and warm conveyor belts. EMLs, or rather Spanish Plumes, are probably the most thoroughly studied special cases of ADs in Europe (e.g. Schultz et al., 2025b; Carlson and Ludlam, 1968; Lewis and Gray, 2010; Dahl and Fischer, 2016). However, the air stream that behaves EML-like is not so frequent and large parts of Europe experience ADs much more frequently than the British Isles which are the main target region for Spanish Plumes (see Fig. 1a). The ascending, warming, drying air stream that is found for almost all AD cases
resembles a warm conveyor belt, but it is much dryer since it originates in northern Africa.




*Code availability.* Much of the data processing is done with the climate data operators (`cdo`; Schulzweida, 2023) and `python`. Code is available at zenodo: Fix-Hewitt (2025)

*Data availability.* ERA5 data is freely available at the Copernicus Climate Change Service (C3S) Climate Data Store (Hersbach et al., 2023). The results contain modified Copernicus Climate Change Service information 2020. Neither the European Commission nor ECMWF
is responsible for any use that may be made of the Copernicus information or data it contains. Lightning data from Blitzortung.org is available as participant of measurement network. The LAGRANTO is available from: Sprenger and Wernli (2015).

*Author contributions.* GM and AZ acquired the funding for this project. FFH conducted the calculations, the analysis, and wrote the manuscript under supervision by GM and AZ, with the support of IS and RS. IS and FFH acquired the data and RS supported in software development. GM, AZ, IS, and RS reviewed the manuscript prepared by FFH.

*Competing interests.* The authors declare that they have no conflict of interest.

*Acknowledgements.* We thank Deborah Morgenstern for setting up LAGRANTO and running the first explorative trajectory calculations. We thank all colleagues who were involved in discussions. The computational results have been achieved [in part] using the Austrian Scientific Computing (ASC) infrastructure. This work of Fiona Fix-Hewitt was funded by the Austrian Science Fund (FWF, grant no. P35780).

## Appendix A: Seasonal Analysis

The distribution of the lowest and highest AD-cells is similar across the seasons (Fig. A1 left and centre columns). Both the lowest and highest AD-cells are highest in JJA (upper bounds up to more than 10 km, Fig. A1e) and lowest in DJF (lower bounds as low as around 1 km on average, Fig. A1j). Also the distance between the lowest and highest AD-cells is largest in JJA (more than 5 km) and lowest in DJF (Fig. A1 right column). While in MAM ADs of several kilometres thickness on average cover almost the entire continental portion of the domain, the other seasons show more localised occurrence of the
thickest ADs. In JJA there is a clear maximum thickness over the Baltic states an the Mediterranean Sea.

The seasonal analysis of the conditional probability of a lid shows that the highest probabilities of a lid are recorded in SON and DJF, with probabilities up to more than 55 % in the Mediterranean and west of Iberia (Fig. A2d,j). The Mediterranean is the region with the highest probabilities in all seasons (Fig. A2 first column). The highest mean lid streak length is observed over the Mediterranean in JJA with more than 3.5 days on average, followed by SON (Fig. A2e,h). The absolute longest persisting
streaks (90th percentile above a week) happen in SON and JJA (Fig. A2f,i).



**Figure A1.** Average altitudes of lowest (left column), and highest (centre column) AD-cells, and their distance (right column) per season (rows).



**Figure A2.** P(lid | AD) (first column), mean lid streak length in days (second column), and maximum lid streak length in days (third column), for each season MAM (top row), JJA (second row), SON (third row), DJF (bottom row).



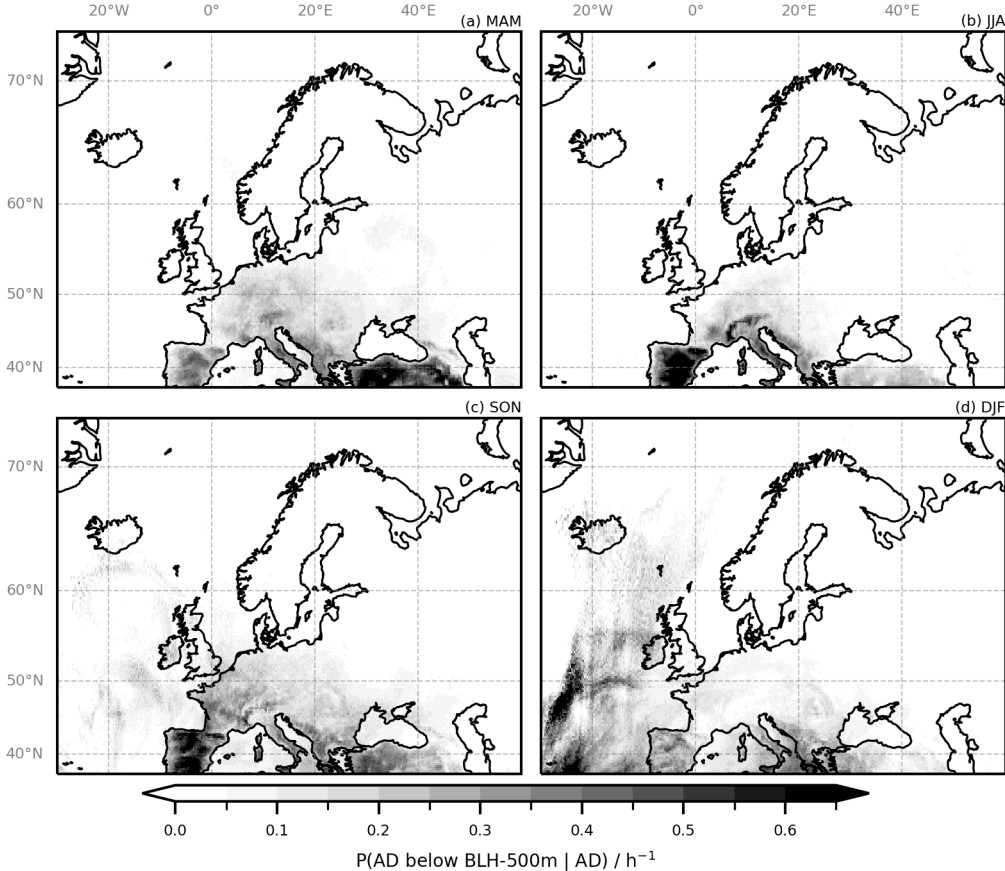

**Figure A3.** Probability of the lowest AD-cell being within the BL, P(AD below BLH | AD), per season.

The highest probabilities for an AD to enter the local BLH are found over mountainous areas, in particular during he warmer seasons, when the local BL is additionally heated. Especially high values are found in JJA and SON over the high regions on the Iberian Peninsula, the Alps, the Apennines, and the Dinaric Alps. In these seasons also the larger Mediterranean islands experience higher probabilities. The south-eastern Mediterranean region has elevated values in MAM already, possibly indi-
cating warm, deep BLH earlier in the year in this region. In autumn, the eastern Mediterranean sea has elevated probabilities, likely because in this season the sea is still warm compared to the surrounding land masses.



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
