# Peer review of "Properties and Characteristics of Atmospheric Deserts over Europe"

_EGUsphere, 2025_

## Referee Comment (RC1)

**Summary & general assessment**

Fix-Hewitt et al. generalize the concept of atmospheric deserts (ADs) with a climatology for AD events over Europe from two years of reanalysis data. They focus on ADs from air masses originating over North Africa that are advected over Europe. They compute a large number of Lagrangian trajectories for such advection events and identify ADs in those. Based on the identified ADs, they assess key properties of ADs, including their frequency of occurrence and spatial extent, synoptic patterns during which they occur, their duration, vertical structure, and typical thermodynamic and vertical pathways.

The presented work is a logical continuation of previous research, and provides the first systematic climatological analysis of ADs for Europe. The methods used are appropriate for the application, and largely well explained in the manuscript. The paper is well structured, however, language in several sections could be improved for a slightly smoother read. Figures are appropriate and well chosen to visualize key findings, however, visualisation could be improved (see specific comments). Overall, I am pleased with the quality of the manuscript and deem the topic well suited for publication in Weather and Climate Dynamics.

**Major comments**

**Title**

I think the title has more potential to highlight the key novelty of the paper: A first climatology of AD events over Europe. Adding the term "climatology" would also make it easier to differentiate from the authors' previous publication that explored ADs with form of a case study.

**Methods**

I acknowledge that many of the deployed methods require the selection key parameter values, and that this is partially based on previous work of the authors. However, I think this still requires more explanation and justification in the presented manuscript. In particular:

- Why exactly do you chose three clusters for the synoptic analysis? What would change with 4 or more clusters?
- Typical trajectories: why do you use a subjective selection and not a systematic one? You could cluster the trajectories and show the cluster mean composite, or the trajectory with the smallest deviation from the cluster mean.

**Connection to other synoptic dynamics**

While ADs are connected with particular weather situations and extreme weather, it would be interesting to see a broader discussion of these. Your synoptic analysis shows a situation that somewhat resembles that of North African cyclones (cluster 1). These are known to be efficient for dust emission in North Africa during spring, and can lead to dust transport towards Europe. References (e.g.):

- Bou Karam, D., Flamant, C., Cuesta, J., Pelon, J., & Williams, E. (2010). Dust emission and transport associated with a Saharan depression: February 2007 case. Journal of Geophysical Research: Atmospheres, 115(D4).
- Barkan, J., Alpert, P., Kutiel, H., & Kishcha, P. (2005). Synoptics of dust transportation days from Africa toward Italy and central Europe. Journal of Geophysical Research: Atmospheres, 110(D7).

Is there a connection between ADs and dust transport? Is there a seasonality of the synoptic situation that causes ADs? Also, a mention of the somewhat related term "atmospheric rivers" (ARs) might be interesting. Both, ADs and ARs are connected to extreme events, even though air masses have very different thermodynamic properties.

**Language**
The paper is mostly well written and the authors use adequate vocabulary. However, some sections would benefit from smoother flow of language and improved syntax.
- The abstract would benefit from better phrasing and consistent usage of the same terms that are later used in the paper. See specific comments.
- While it might be common in German to place the verb in the very end, English is much easier to read if you place the verb early. For example, "Persistence is an important feature of ADs, especially in the context of impacts." reads much easier than "Another important feature of ADs, especially with regard to their consequences, is their persistence.". See some suggestions for improvement in my detailed comments.
- I recommend abstaining from using inconcise language such as: almost all, seems, should. You can refer to the numerical values, or clearly state the definition or hypothesis.
- Trajectories. To my understanding, what you calculate forward in time is the location of the air parcel; the trajectory is the path that the air parcel takes/has taken. In the manuscript, the term "trajectory" is used for both. Please rephrase where necessary.

**Figures**
Some figures are hard to "read" due to the choice of colourmaps or feature colours.
- Multi-color but perceptually uniform colourmaps such as cividis or viridis can help to improve the details visible in the plots, while being colorblind-friendly and reproducing well in greyscale. You could further add contour lines to visually highlight the key values that you mention in the manuscript. Black background features such as coastlines are hard to see of you plot fields in dark grey on top of it. Consider changing colours.
- Colourbars use the same colour for low and high and the respective extensions. Either remove extensions or change colours.
- For consistency, I further suggest aligning the alphabetical plot labels (a, b, c, …) to a common location for all plots (e.g., the top left).

Also see specific comments on figures.

**Specific comments**

l. 2: Language consistency: here you write "*between* May 2022 and April 2024", later you write "*through*". Please use consistent term.

l. 3: "at every hour" – I assume you mean "every full our"?

p. 3f: Please add which atmospheric dataset you use for calculating the trajectories (→ ERA5).

l. 5ff: I suggest to reorder the first two paragraphs slightly: First paragraph - introduction of atmospheric deserts and where they occur. Second paragraph - what you do (track,technical details, etc.).

l. 9: "Typically, atmospheric deserts persist for about one day on average". Typically or on average? Suggestion: On Average, atmospheric deserts ...

l. 12: "resides between the local boundary layer height and the troposphere". I do not understand where this is. The top of the boundary layer usually marks the transition to the free troposphere. Please clarify.

l. 13f: "The atmospheric desert air rarely enters the local boundary layer, and if it does, it happens at over...". This sentence does not read smoothly. Suggestion: Rare intrusions of desert air into the boundary layer mainly occur over high orography and during the warm season.

l. 15: "lid" – consider briefly mentioning what this is.

l. 34: Reference: Schultz et al. (2025a). Why is this publication singled out? It overlaps with the previous sentence and references: origin of Spanish Plume air often further south that Iberian peninsula; in North Africa.

l. 36: "by e.g." – remove "by"

l. 40: References. How are these ordered? I suggest sorting by year of publication.

l. 49 "for a few days" – for how long exactly? Use numerical value.

l. 50: "pathways of the trajectories" - pathways or trajectories?

l. 52: "almost preserving its thermodynamic properties and rising only slightly," preserving/rising relative to what? I assume: relative to the thermodynamic properties of the air parcel at the start of the trajectory. Please clarify.

l. 48ff: Inconsistent use of present and past tense. Please stick to one tense in this section when you describe the results from Fix at al. 2024.

l. 59: "what area they span". The following paper analyses what *distances* ADs *span* and what *areas* they *cover.*

l. 61-70: This section would benefit from a brief introduction, before saying which subsections cover what. I suggest reordering, e.g., you could start with the sentence "The focus period of this study…", then say "The direct detection method requires the calculation …", then "We use the ERA5 reanalysis…".

l. 67: Why are computational requirements mentioned here? I suggest removing, moving to the discussion, or adding a reason why you need this info here.

l. 74: "1 hPa". Here and elsewhere: There should be a narrow space between value and unit.

l. 77: Please reference figure 1 which shows the domain(s).

l. 77: **N**orth Africa but **n**orthern Africa

l. 82: "various contexts" - In which contexts exactly? You add four references; please also give examples what these publications use LAGRANTO for.

l. 84: "Naturally" – Why is North Africa "naturally" the source region of interest? Please add.

l. 91: "without affecting the results as tested for that case study". I do not understand this sentence. Please rephrase.

l. 92 "but for this study" – Which study do you mean here? The here-presented study? Please clarify/rephrase.

l. 94: "Per definition AD air should…". A clear definition would say "AD air originates in..." - not "should".

l. 95: "the BL can grow to". Suggestion: The BL can reach several km in depth.

l. 96: "It can be assumed" - Why can this be assumed? Please add reasoning.

l. 99f: What happens to air parcels that are first advected outside of the "curtain" and only then make their way to Europe? What about air from other source regions? It would be good to highlight more that you only focus on air that is advected from North Africa (where we assume most ADs to originate).

l. 108: no need to use parenthesis

l. 112: "decent vertical resolution", "high enough numbers". What do you mean by this? What criteria does the vertical resolution need to fulfil to be "decent"? High enough for what? I suggest to rephrase, give a reason, and add a numerical value (e.g., minimum numbers of trajectories per box),

l. 129: "2 days" – Why this threshold? Please give a reason.

l. 130: "geoh500" – "z500" is a more common notation.

l. 134ff: Is this the "elbow method"? Does k=3 fulfil the previously explained criterium regarding the total sum of squares? What would change with k=4 (or greater)?

l. 140: "on one day" - On the same date?

l. 145: Why 4 clusters? You say "as in Fix et al. (2024)", but please also give a brief reason in this paper why you choose this value.

l. 147: "5th day before each of the maxima (as this is the trajectory length)". This sentence is quite hard to understand, particular which duration of the trajectories you mean here. The information in parenthesis does not particularly help here. Please rephrase.

l. 148: "identified subjectively". Why is the representative trajectory not chosen analytically? For example, you could cluster all trajectories and compute composites, or chose the closest trajectory to each cluster centre.

l. 157: "not without problems". Problems with/of what? Please give relevant examples.

l. 172: "probability". Probability of/for what? I think you mean for an AD being present over the time series analysed here. Please add.

l. 173: "weighted by grid cell area". Even though you might use an area-weighted average for computation of the mean field values, I find this explanation quite confusing here. Figure 1 shows AD probability per hour and area (not per cell), and does not need further weighting for interpretation (I think).

l. 176: Notation "0.7**x**10". Here and elsewhere: this looks line the letter x. Better use the correct "times" symbol, in LaTeX this is \times .

l. 178: "non-weighted". It is not entire clear why you use weighted and unweighted probabilities here. Please add an explanation.

l. 181: "ADs cover between…". You prominently introduced the frequency of ADs at the beginning of the section. For consistency, I think it would be more clear if you state that the second paragraph is about spatial properties of ADs.

L 183: "is smaller". Smaller than what? Please add what quantity or season you refer to.

l. 187f: "especially in the eastern Mediterranean, over Turkey and the Black Sea". What about the Balkans? Figure 1d) suggests that at least the southern Balkans are equally affected.

l. 189: "noticeable". What does noticeable mean here? Better mention some numerical values.

l. 190: "20%" - 20% of what?

l. 198: "does show that it is rare that no AD air is present in the domain north of 37∘ N". It would be interesting to know how often this occurs over the analysed period.

l. 206: "during a fully developed AD-event". Or rather leading to a fully developed AD event?

l. 209: "shown". If you mention that features are shown somewhere, please also mention in which figure (here: reference figure 3).

l. 212: "typical trajectory cluster average paths". Suggestion: "for the typical trajectory for this cluster." Also, it would be interesting to see how this "typical trajectory" looks like. Consider adding this to the supplementary material.

l. 218: "trajectories being transported". Air is transported, trajectories are located. Please rephrase.

l. 229: Fig. 4: Reference this figure where you first describe streak lengths (previous paragraph).

l. 233: "closer to a week in the Mediterranean in SON". Figure 4 shows less than one week (7 days) in the Mediterranean for SON. Please clarify.

l. 235: Punctuation: "This indicates, that". No comma.

l. 239: Language: "ADs seem to be". Are they or are they not?

l. 242: "distribution". Suggestion: spatial structure/pattern

l. 245: "The largest average distance between the highest and lowest AD-cells". Suggestion: The largest vertical depth …

l. 251: "cell-hours". What are cell-hours? Please introduce and explain.

l. 251f: "South of 55∘ N and west of 40∘ E […] . Beyond this [...]". The former also includes the domain that was mentioned in the previous sentence and which shows a lid in 45% of daytime hours. Please rephrase. The latter could be rephrased to "In other parts of the domain".

l. 265: "compared to a continental one, as it is often cooler and moister". What does this mean for the development and persistence of a lid? Please add/explain.

l. 272: "simply due to its origin". Origin in where? Please add/explain.

l. 273f: Any estimate how reliable ERA5 reproduces actual BLH, particularly in mountain regions?

l. 274: "entering" – intrusion?

l. 293f: You already said in the previous sentence that figure 9 is a schematic. You can add the reason that no scales shown to the figure caption.

l. 297f: "another favoured altitude is between 7 and 8 km". I do not see how this band stands out compared to lower layers. There is not even a local maximum here. However, AD cells can frequently be found in up to about 10 km. Please rephrase or explain why/how you identify this band as particularly prominent.

l. 298: "medium altitudes". Which altitude do you mean with "medium"? Please add numerical values.

l. 299 "significantly". Did you do any significance testing? If not, better avoid this term.

l. 306: "At least one ... almost .. all". This is quite blurry language, please rephrase.

l. 310: Punctuation: "trajectories that end up at lower altitudes, cooled considerably". No comma.

l. 311: "fits well with cluster d (Fig. 9d, red), which also appears in many cases". I think cases appear within the clusters, not the other way round.

l. 315: Based on Fig. 8c, I struggle to see how q remains almost unchanged between 2 and 6 km. Or do you mean Theta_E?

l. 323f: "few degrees". This appears to be a temperature unit and should therefore say "Kelvin". However, since you do not show any scales in your schematic, I suggest not referring to absolute values that cannot be seen in the figure.

l. 329: "due to distance to the source". You might want to add that this is due to the longer advection time that is required to reach these locations.

Section 4. The section title promises an outlook, but I rather see a discussion with conclusions and an abrupt end. Has the outlook gone missing?

l. 351f: It would be interesting to see a seasonality analysis for the different synoptic clusters. E.g., cluster 1 aligns well with the synoptic situation we see during North African Sharav cyclones. These are known to be an important driver for dust emission during spring. Cluster 2 resembles frequent synoptic conditions during events of Saharan dust transport towards Central Europe. ADs might have an impact on predictability not only due to their thermodynamic properties, but also due to their potential aerosol load.

l. 359: "almost not happening". Suggestion: "rarely happens (<20%)"

l. 385f: I did not see any lightning products in the presented figures nor any mentions in the manuscript. Potential copy/paste from previous paper?

l. 465ff: Remove link to Google books.

**Specific comments on Figures**

**Figure 1**
- Labels a, b, c, … I suggest to align locations, e.g., to top left.
- Colourbar: Extensions seem to have same colour as min/max field. Either remove extensions or chose different colours.
- Colourmap: While it is okay to use Greys, please use a different colour for the map features (coastline).
- Caption: You mention cells, but your colourbar legend shows area im km². Please clarify.

**Figure 2**
- I suggest to start (and) the x-axis tick labels with the boundaries of the time period that you analyse.

**Figure 3**
- Coastlines have same colour as high values of AD frequency. This makes it hard to locate the extreme values in cluster 3. Please change coastline colour/ AD frequency colourmap.
- What unit is AD frequency?
- It would be very interesting to see a seasonality analysis of these clusters.

**Figure 4**
- I find the maxima very hard to see. You could highlight key thresholds with contourlines.
- Consider choosing a different colour for the coastline, or the 90th percentile fields.

**Figure 5**
- I find it very hard to see clear structures in here. You could use another colourmap such as viridis or cividis (equally perceptually-uniform) that pop out more gradual changes in the fields. Consider adding contour lines for highlighting the values you mention in the manuscript.

**Figure 9**
- I think it is enough to mention once that this is a schematic. You can add "curves are relative and not to scale".

**Figure A1**
→ see comments for figure 5

---

## Referee Comment (RC2)

**Review of WCD-2025-3552: Properties and Characteristics of Atmospheric Deserts over Europe**

**Overview:** This manuscript quantifies and extends the work of Fix et al. 2024 on 'atmospheric deserts' (ADs) over Europe using the ERA5 reanalysis from 2022-2024. They explore the frequency of occurrence of these events through meteorological seasons, as well as other properties including layer depth and thermodynamic evolutions. They also explore clusters of common weather patterns associated with ADs, and provide a subjective clustering of common trajectories that transit from Norther Africa into Europe and the Mediterranean. I found the paper interesting and a nice extension of the prior work, though I felt that several opportunities to connect ideas within the paper and to our broader understanding were either under-discussed or missed. This said – I believe the paper will be a nice addition to WCD upon revision.

**General Comments:**

1. *Overall structure and grammar:* There were a few instances where the formatting of the paper or sentence structure got in the way of interpretation. In several instances the author's opted for a series of short paragraphs (1-3 sentences) where a single, more connect, paragraph was a better fit. Specific areas where this was noted include the abstract (which should really generally be a single paragraph), the methods, and the conclusion/discussion. There were also other instances where the sentence structure muddled the author's point (I tried to point out a few below in the specific comments). Lastly, please be careful with word choices. There were instances where the choice of word misconstrued the point of the sentence (for example in the abstract 'The third air stream in this group is the previously known …' – was this previously known as this and now has another name? Or is it commonly known as?).

2. *Figure clarity:* I appreciate the authors frequent use of black and white color bars (which work well for a variety of users). However, the choices also made it challenging to pick out certain details, as the lightest shading of grey was too close to white, and the darkest shading made it impossible to make out geographic borders (eg. Fig. 1). The color bar choice for Fig. 5 (a,b) also was really hard to interpret – perhaps aim for one with a sharper gradient across the data range. Lastly – though this may be a personal choice of the authors, I found the presentation of axis/colorbar labels as 'variable / unit' potentially confusing (it briefly made me think you were dividing the two). I would adjust to a style of 'variable (unit)' intead to avoid this confusion.

3. *Clustering analysis:* I was admittedly a bit surprised to see that of the three synoptic clusters that you selected, ~60% of the cases fell in a single cluster. I was happy to see the that you applied a total sum of squares approach, but when expanding the number of clusters, did you see additional differences that were of interest? Cluster 1 looked a bit like it was getting zonally smoothed (perhaps zonally smoothed anticyclonic Rossby wave breaking), so I was curious if this was the case. As a follow-on to this, it would be interesting to either a) do additional cluster analysis by season (perhaps for supplemental figures) or to provide a breakdown of the seasonal composition of each cluster (i.g. what percent of each cluster population came from each season). Regarding the trajectory clustering, did you attempt any sort of automated technique to cluster rather than a subjective selection? You may instead wish to present these as representative pathways and perhaps show them on a map. Lastly regarding the trajectory 'clusters' – how did these 6 different clusters project onto the 3 synoptic clusters?

**Specific comments:**

- Line 3: I'd avoid the word 'suggested' when discussing literature when possible. Please also remove the 'by' before 'eg.').
- Line 49: Please change the word 'the' to 'an' after June 2022.
- Lines 81-82: Please remove 'in atmospheric science, in various contexts'.
- Line 125: I found the sentence starting on this line ('Then, local maxima …') unclear as written.
- Line 132: I wasn't sure why it was mentioned that the data was reshaped – had it previously been unshaped? You can probably just remove this sentence.
- Lines 134-135: I found the two sentences here ('The suitable number … with additional clusters') hard to interpret. Consider expanding – in particular because the choice of clusters can be critical to the ensuing analysis.
- Line 156: Please include an 'is' between 'suggest it' and 'by several'.
- Line 161: Did you consider allowing the daytime hours definition to vary by season? It seems like for much of this reason, you're cutting off daytime hours in the summer by early to midafternoon when the BL may still be deepening.
- Figure 1: I found the ordering of panels here a bit odd. I would aim to go in chronological order for panels d-g (eg. MAM, JJA, SON, DJF) or similar.
- Lines 195-196: Do we know that a greater poleward extent equates to a longer duration? This seems counter to some of your other results. I would think the poleward extent vs. duration would highly depend on the synoptics that got the air to high latitudes in the first place (eg. a strong cyclone vs. a moderate anticyclone will have different advective velocities).
- Line 215: I'm not sure 'intensifies strongly' is the wording I would go for here. Perhaps amplifies?
- Line 215: For the statement 'this pattern likely …' – perhaps the word 'typically' instead? You've likely analyzed this to know the answer here and could more strongly say 'typically'.
- Lines 211-221; Figure 3: Aim to include panel labels here to clarify the discussion. In addition, I would consider either creating a second figure (or adding to figure 3) a composite of MSLP or 850 hPa heights here to capture the lower tropospheric flow. This matters both for the 24h after onset period (where the lower troposphere presumably has done much of the initial advection of the AD from Africa) and the latter period (which matters for your lower tropospheric trajectories). Consider including this and expanding/support the discussion accordingly.
- Lines 223-229: Can these 'streaks' be reinforced by multiple synoptic scale events? It seems like a residence time of >7 days needs multiple events given the atmospheric variability in this region. As such, how does this impact the interpretation of your clusters?
- Line 233: Please change to '…below a week, but is closer to …'
- Line 243: I'm not sure I can see this north-east tilt from your figures. Also, given the spatial dimensions and rate of ascent here, how much of a horizontal extent of tilt should we expect to see?
- Line 247: Please remove 'in their centres'.

- Lines 256-267/methods: I found this definition of the lid challenging to feel 'comfortable' with. Why not go for a more rigorous definition like exceeding a threshold for vertical gradient in potential temperature? You have the data (given your later figures), and one would expect a larger magnitude vertical gradient in theta for a profile with an AD over the BL rather than one with a common free atmosphere overhead.
- Lines 256-258: Would it be better to show the fraction of the mean/90$^{th}$ percentile streak length that met the criteria of a lid instead? I feel like this may be a more natural extension/connection of the work.
- Lines 268-269: The sentence structure here is off – aim to flip the order of discussion here.
- Lines 274-277: Aim to include discussion of issues with resolving the BL in these areas of complex terrain as well. I suspect this plays as much, if not more, of an issue.
- Line 279: Please flip the order of 5 and 1 here.
- Lines 278-282: Though heat waves matter here, please also consider adding discussion here on this being important for suppressing deep convection as well.
- Lines 292-295: Is there a way for you to normalize the data (rather than strip the units) here to solidify the analysis?
- Line 303: Careful on the term 'warmed on average'. An increase in potential temperature does not necessarily equate to an increase in temperature.
- Line 303: Please add a space between g and kg$^{-1}$.
- Line 315: I'm not sure I see this 'unchanged' characteristic, in particular for q.
- Figure 8a: Please include the vertical axis here too.
- Section 3.5: Two points here to consider. First, I felt like theta-e (an excellent metric for an air mass) was presented but not really discussed. I think one key take-away here is that theta-e isn't really changing much (perhaps a slight weak bias toward a negative tendency), indicating that once the air mass is being advected, it's broad airmass characteristics are not really changing. In other words, changes to the moisture characteristics are being compensated by changes to the dry parcel characteristics, resulting in a nearly-conserved air mass overall property. The second point here (noted in the general comments above) – it would be really helpful to see how your trajectories here relate to the 500 hPa composites. There are a variety of ways to approach this, but I think it would really help connect the different analysis approaches of the manuscript well.
- Figure 9: Please clarify what panels a-f are a bit more clearly.

---

## Author Comment (AC1)

**Answer letter to Reviewers**

Dear reviewers.

thank you for your thorough and constructive feedback. Both of you raised very valid concerns and we are happy to incorporate your feedback to improve our manuscript.

In the following, we address your comments individually. In order to avoid repetition, we grouped your comments by topic and answer them together. Please find the comments by *Reviewer 1 in italic*, those by **Reviewer 2 in bold**, and our answers in normal font. Major comments are addressed before Specific ones and Figure-specific ones.

All the best, Fiona Fix-Hewitt and Co-authors

**Major Comments**

I think the title has more potential to highlight the key novelty of the paper: A first climatology of AD events over Europe. Adding the term "climatology" would also make it easier to differentiate from the authors' previous publication that explored ADs with form of a case study.

Thank you for this suggestion. We originally refrained from using the term 'climatology', since the dataset is only 2-years long. We do see your point about differentiating our two papers more, and decided for "Properties and Characteristics of Atmospheric Deserts over Europe: A First Statistical Analysis".

Language and Figures:

The paper is mostly well written and the authors use adequate vocabulary. However, some sections would benefit from smoother flow of language and improved syntax. [...]

Some figures are hard to "read" due to the choice of colourmaps or feature colours. [...]

There were a few instances where the formatting of the paper or sentence structure got in the way of interpretation. [...]

In several instances the author's opted for a series of short paragraphs (1-3 sentences) where a single, more connect, paragraph was a better fit. I appreciate the authors frequent use of black and white color bars (which work well for a variety of users). However, the choices also made it challenging to pick out certain details, [...] Lastly – though this may be a personal choice of the authors, I found the presentation of axis/colorbar labels as 'variable / unit' potentially confusing (it briefly made me think you were dividing the two). I would adjust to a style of 'variable (unit)' instead to avoid this confusion.

We thank the reviewers for their comments on improving the readability of our manuscript and will incorporate the suggestions as fully as possible. Colour maps are adjusted accordingly, i.e. the colour range has been adapted, so that the grey scale for example does not reach the very light and very dark colours, which improves readability. While it was a personal choice, we adjusted the axis descriptions as suggested to avoid confusion.

Regarding the length of individual paragraphs, we will carefully assess the matter but we also think that this is personal preference and style. In particular, we do want to keep several paragraphs in the abstract, as they separate different ideas.

I acknowledge that many of the deployed methods require the selection key parameter values, and that this is partially based on previous work of the authors. However, I think this still requires more explanation and justification in the presented manuscript.

Synoptic Analysis

Why exactly do you chose three clusters for the synoptic analysis? What would change with 4 or more clusters? Is there a seasonality of the synoptic situation that causes ADs?

I was admittedly a bit surprised to see that of the three synoptic clusters that you selected, 60% of the cases fell in a single cluster. I was happy to see the that you applied a total sum of squares approach, but when expanding the number of clusters, did you see additional differences that were of interest? Cluster 1 looked a bit like it was getting zonally smoothed (perhaps zonally smoothed anticyclonic Rossby wave breaking), so I was curious if this was the case. As a follow-on to this, it would be interesting to either a) do additional cluster

analysis by season (perhaps for supplemental figures) or to provide a breakdown of the seasonal composition of each cluster (i.g. what percent of each cluster population came from each season).

In response to these comments we did look into the clusters more closely and we agree, Cluster 1, which has most of the cases indeed smooths various different situations. We therefore refined the clustering method further. Instead of normalising the individual maps, we use standardised anomalies of  $Z_{500}$ . This is done by subtracting the mean and dividing by the standard deviation, using a 30-day running window. This helps to emphasise anomalous patterns and to avoid that a generic 'westerly' current is picked up by the clustering.

We already used a principal component analysis (PCA) in the manuscript, but we refined its use further. Using PCA reduces the dimensionality of the data, but it also helps identifying the spatial patterns repsonsible for the most variance in the data. We find that using the first 7 principal components (PCs) is a good compromise between retaining most of the variance (62%) and simplifying. Plotting the PCs against their explained variance ratio also shows that taking further PCs into account can increase the explained variance ratio only by small amounts ('elbow', see Fig. R1). However, the results are quite robust the number of used PCs. The PCs are not normalised before clustering by them, since we want to take advantage of the emphasis they give to the important spatial patterns in the data (which also makes it more stable towards the number of PCs used in the clustering).

Finally the data is clustered. With the refined method, 4 clusters is the better choice. This is based on the elbow plot of the total sum of squares (see Fig. R2) and a judgement of the interpretability of the resulting cluster mean patterns. However, the overall patterns are similar for 3, 4, and 5 clusters, as there is a more zonal cluster, one with a trough and then some constellations of ridges - we therefore believe 4 clusters is a useful number of clusters to interpret. In order to show some more context, we decided to also show the observation that is closest to the cluster centroid (Fig. R3), instead of just the cluster mean. This also allows to show trajectory clusters together with the synoptic situation.

Dividing the 66 AD-events by season results in only 15, 19, 16, and 16 events per DJF, MAM, JJA, and SON, respectively. Clustering these will yield clusters with very few members per cluster, which we think is not useful for interpretation. A breakdown of the seasonal composition of the clusters, however, it more useful. Hence, for each cluster we show the percentage of events from the respective seasons (Fig. R4). As you can see, all clusters have members from all season. Cluster 1 has less members from winter, only few events from summer fall into Cluster 2 and Cluster 4 has only few members from autumn. The figures and discussion will be added to the manuscript.

Typical trajectories

why do you use a subjective selection and not a systematic one? You could cluster the trajectories and show the cluster mean composite, or the trajectory with the smallest deviation from the cluster mean.

Regarding the trajectory clustering, did you attempt any sort of automated technique to cluster rather than a subjective selection? You may instead wish to present these as representative pathways and perhaps show them on a map. Lastly regarding the trajectory 'clusters' – how did these 6 different clusters project onto the 3 synoptic clusters?

We acknowledge that the selection of the typical trajectories can be confusing. In the revised manuscript we adopt a different, and more objective approach. To clarify: We are already clustering the trajectories per starting day. Clustering the cluster averages again, is going to be very difficult, because while they do show similar behaviours, their path on the map and the timing and magnitude of changes in altitude or thermodynamic variables can vary strongly. Therefore, we define the different air stream types by using thresholds:

blue  $\Delta h$ . a.m.s.l > 1000 m &  $\Delta \theta$  > 0 K &  $\Delta q$  < 0 gkg-1 red  $\Delta \theta$  < 0 K &  $\Delta q_{rel}$  > 0.1 gkg-1 yellow  $\Delta \theta$  < 0 K &  $\Delta q_{rel}$  < -0.1 gkg-1 green  $\Delta \theta$  < -1.5 K &  $|\Delta q_{rel}|$  < 0.1 gkg-1 cyan  $|\Delta \theta|$  < 1.5 K &  $|\Delta q_{rel}|$  < 0.1 gkg-1

Here,  $\Delta$  refers to the difference between the end and start of the cluster-averaged trajectory.  $\Delta q_{rel}$  denotes the relative change in q since that time. An absolute relative change in q of less than 10% over

several days is used as a criterion to identify an air stream that retains its moisture. Similarly, a change in potential temperature of less than 1.5 K is considered small. Clusters that do not meet the criteria for any of these air stream types are categorised as black. The criteria were selected based on visual inspection of the clustering results and because the resulting typical air streams can be linked to physical processes that dominate the modification of the air during its advection: The blue air stream implies latent heating due to condensation and a loss of humidity caused by precipitation,. The red air stream may be influenced by evaporation or mixing with cooler, moister air masses. The yellow air stream indicates mixing with cooler, but dryer air masses, the green one is dominated by radiative cooling, and the cyan one behaves like the previously studied EMLs.

For each of the 66 AD-events, we perform clustering of the trajectories. Specifically, we consider trajectories initialised 3 and 5 days before the AD maximum (i.e. for a maximum at 2022-06-20 08:00, trajectories initialised on 2022-06-15 and 2022-06-17 are clustered, respectively). This results in  $66 \times 2 \times 4 = 528$  clusters. For simplicity we do not use all days between the onset and maximum any more, but only two starting days per event to capture different constellations.

Instead of displaying a schematic Figure in Fig. 9, we display the cluster means of 2 cases, that cover the different air stream types. The results for all cases will be shown in the appendix, for completeness. The breakdown of how often which air stream is found, as well as the seasonal distribution will be discussed in more detail.

Connection to other synoptic dynamics: While ADs are connected with particular weather situations and extreme weather, it would be interesting to see a broader discussion of these. Your synoptic analysis shows a situation that somewhat resembles that of North African cyclones (cluster 1). These are known to be efficient for dust emission in North Africa during spring, and can lead to dust transport towards Europe. [...]

Thank you for mentioning relevant literature. We add some context to the discussion, referring to some more recent work (partly based on the ones you suggested) about warm air intrusions and dust events: "The identified flow patterns also resemble those identified in other studies in the context of Saharan warm air intrusions and dust transport. Cos et al. (2025) also found anomalously low pressure in (north-) western Europe during the onset of warm air intrusions from the Sahara into the Mediterranean. Varga et al., (2013) identified different types of geopotential height patterns connected to Saharan dust events in the Carpathian Basin. Their Type I has similarities with our Cluster 1. Their Type II resembles our Cluster 2, and is also most frequent in spring. Their Type III also has a ridge over central Europe similar to our Cluster 3, which causes dust events especially in western Europe. Also Rostási et al. (2022) investigate dust intrusions into central Europe, and several of their situations resemble our clusters as well. These similarities can be expected, as both warm air as well as dust intrusions must be caused by ADs by definition."

Rostási, A., Topa, B. A., Gresina, F., Weiszburg, T. G., Gelencsér, A., and Varga, G.: Saharan Dust Deposition in Central Europe in 2016—A Representative Year of the Increased North African Dust Removal Over the Last Decade, Frontiers in Earth Science, 10, https://doi.org/10.3389/FEART.2022.869902/FULL, 2022.

Cos, P., Olmo, M., Campos, D., Marcos-Matamoros, R., Palma, L., Ángel G Muñoz, and Doblas-Reyes, F. J.: Saharan warm-air intrusions in the western Mediterranean: identification, impacts on temperature extremes, and large-scale mechanisms, Weather Clim. Dynam, 6, 609–626, https://doi.org/10.5194/wcd-6-609-2025, 2025.

Varga, G., Kovács, J., and Újvári, G.: Analysis of Saharan dust intrusions into the Carpathian Basin (Central Europe) over the period of 1979–2011, Global and Planetary Change, 100, 333–342, https://doi.org/10.1016/J.GLOPLACHA.2012.11.007, 2013.

Is there a connection between ADs and dust transport?

There is indeed a connection between ADs and dust transport from the Sahara. We have briefly investigated the conditional probabilities of dust aerosol optical depth (*duaod*550, based on EAC4 reanalysis) anomalies greater than 1, given the presence or absence of AD air (Fig. R5). This analysis

yielded that high duaod550 in absence of AD air is very uncommon – which logical, since dust must be picked up in the desert boundary layer, therefore the air transporting it is per definition AD air. The low conditional probabilities that do exist are likely due to the limited length of the trajectories, which are no longer tracked after 5 d. On the other hand, we find that up to 50 % of AD-hours over continental Europe also have anomalously high duaod550. Hence, when dust is transported towards Europe, it is most certainly due to an AD, and in up to half of the time, ADs are dusty. Since adding a lot of detail about consequences of ADs to this manuscript is beyond its scope, we have further work planned on the consequences of ADs. That will include this and more detailed analyses. However, we mention it in the current work briefly for clarification: "Another consequence of ADs is the transport of Saharan dust. By definition all dust events need to be ADs, but not all ADs have to carry dust. The question of whether and how often they do is beyond the scope of this study."

Also, a mention of the somewhat related term "atmospheric rivers" (ARs) might be interesting. Both, ADs and ARs are connected to extreme events, even though air masses have very different thermodynamic properties.

The name atmospheric deserts is indeed inspired by atmospheric rivers and both are connected to extreme events. We add: "Atmospheric rivers are another phenomenon related to extreme weather events. While they sound like the moist counterpart of ADs, the direct comparison is not straightforward, as atmospheric rivers are defined by their water vapour footprint, while ADs are solely defined by their source region."

**Specific Comments**

Specific comments that are specific to language, sentence structure, or similar, and do not need clarifying comments, are not answered here in detail, but incorporated in the revised manuscript directly.

l. 12: "resides between the local boundary layer height and the troposphere". I do not understand where this is. The top of the boundary layer usually marks the transition to the free troposphere. Please clarify.

Thank you for making us aware, this is indeed misleading. We will clarify as: "Atmospheric desert air frequently resides above the local boundary layer and extends through much of the free troposphere."

Since there is no clear definition used in that paper of when the AD event starts and ends, nor is it defined more closely what 'a large part of Europe' is, we do not think that adding a numerical value here will give clarity.

The computational resources are mentioned here, to explain why the investigated time period is only 2 years long, rather than covering much longer (climatological) time scales. We move this to the discussion.

Figure 1 does not show the entire domain used in the calculations, which reaches as far south as 15° N, hence it would be misleading to refer to it. We also do not want to extend the shown domain in Fig. 1b to capture the entire domain, as it would reduce readability, and there is no necessity to show the entire data domain. We need the data domain to reach this far south, to capture the entire possible source region south of the curtain, in case trajectories re-circle before reaching Europe.

Changes are made according to the comment of Reviewer 2, since it is not relevant for this study, what the other studies used LAGRANTO for specifically.

l. 49 "for a few days" – for how long exactly? Use numerical value.

Since there is no clear definition used in that paper of when the AD event starts and ends, nor is it defined more closely what is large part of Europa' is used a not think that adding a numerical value have

l. 67: Why are computational requirements mentioned here? I suggest removing, moving to the discussion, or adding a reason why you need this info here.

l. 77: Please reference figure 1 which shows the domain(s).

l. 82: "various contexts" - In which contexts exactly? You add four references; please also give examples what these publications use LAGRANTO for.

Lines 81-82: Please remove 'in atmospheric science, in various contexts'.

l. 99f: What happens to air parcels that are first advected outside of the "curtain" and only then make their way to Europe? What about air from other source regions? It would be good to highlight more that you only focus on air that is advected from North Africa (where we assume most ADs to originate).

I am not entirely sure if I understand your question, but I will try to answer to my best knowledge. We are considering all air parcels that have a northward component at the curtain. There are only two ways an air parcel that is not considered reaches Europe later on: 1) it starts travelling south/'inside' the curtain and turns around at a later time 2) it travels 'outside' of the curtain, but without a northward component. In the first case, that parcel will be captured at a later time, when it does pass through the curtain with a northward component. The latter case is not likely, as it requires almost purely zonal movement due to the orientation of the curtain. That also means, that those parcels likely travel over the Atlantic before eventually possibly reaching Europe. We do not want to capture these particular parcels, since they might have interacted with the marine BL there and altered their properties, so that they cannot really be considered AD air any more when they reach Europe.

We highlight more clearly that due to its size North Africa is the dominant source region for Europe and that we only focus on this region: "In this work, we are interested in ADs over Europe. Naturally, North Africa is the source region of interest, since it is the biggest desert area in the vicinity. For the case study in Fix et al. (2024) it was shown that the proportion of trajectories originating in Iberia is very small, and literature also suggests, that often the air involved in Spanish Plumes is actually of subtropical origin (e.g. Schultz et al., 2025a, b), therefore we neglect Iberia as an additional source region. Trajectories are initiated along a 'curtain'..."

l. 112: "decent vertical resolution", "high enough numbers". What do you mean by this? What criteria does the vertical resolution need to fulfil to be "decent"? High enough for what? I suggest to rephrase, give a reason, and add a numerical value (e.g., minimum numbers of trajectories per box),

With a vertical resolution of 500 m, the mean trajectory count per cell is 6.8, and only 18.7% of the AD-cells over the entire domain and period have counts of greater than 10. Hence, increasing the vertical resolution will result in very low trajectory counts per cell, which reduces robustness. We do not want to decrease the vertical resolution further, since that would mean we cannot resolve features of scales smaller than 1 km any more.

The numbers of trajectories per cell already are quite low - especially also compared to those in our previous study (Fix et al. 2024). This is due to a reduction of trajectories in general to make the method computationally feasible for longer time periods. We re-calculated the case from that study, and find that the results remain qualitatively the same, so we believe that the lower number of trajectories does not cause fundamental problems.

Adjusted in the manuscript: "In order to identify the AD air mass, the trajectories are aggregated to grid boxes of  $0.25^{\circ} \times 0.25^{\circ} \times 500 \,\mathrm{m}$ , matching ERA5 grid cells in the horizontal. A higher vertical resolution would reduce robustness of the analyses as it would decrease the number of trajectories per AD-cell (which is 6.8 on average for  $500 \,\mathrm{m}$  resolution). At a lower resolution, we could not resolve features smaller than 1 km properly any more. Hence,  $500 \,\mathrm{m}$  is chosen as vertical resolution. An AD-cell is then defined as a grid box that contains at least one trajectory. This results in a dataset that designates each point in space and time as AD- or nonAD-cell. Additionally, the average properties of all trajectories within that cell are known. Using only one trajectory as threshold to identify a cell as AD-cell may seem like a weak definition, but as argued in Fix et al. (2024) it is a useful one and was shown not to substantially misidentify the AD-cells."

l. 129: "2 days" - Why this threshold? Please give a reason.

We added a clarification to the manuscript: "The situation 24 h after the minimum and 24 h before the maximum are taken as representative for the early and mature phases of an AD-event respectively. To ascertain that they do not overlap, only those 66 events with at least 48 h between their minimum and maximum are taken into account for further analysis."

**Line 132: I wasn't sure why it was mentioned that the data was reshaped – had it previously been unshaped? You can probably just remove this sentence.**

For the sake of reproducibility we will keep this sentence but try to clarify: "The data is then reshaped from 3-dimensional format (lon  $\times$  lat  $\times$  time) to 2-dimensional format (space  $\times$  time). This reshaping is necessary as principal component analysis (PCA) requires the data to be in (features  $\times$  samples) format."

Line 161: Did you consider allowing the daytime hours definition to vary by season? It seems like for much of this reason, you're cutting off daytime hours in the summer by early to midafternoon when the BL may still be deepening.

We appreciate this comment. We want to make clear at this point that we do not chose daytime hours here to capture the entire daytime BL, but rather because we know we can trust the ERA5 BLH during this time more than at at night. In response to your comment, we redid the calculation several times, to investigate how sensitive the results are to the choice of daytime hours. The following experiments were conducted: 1) 12 UTC only, 2) 10-17 UTC allyear, 3) seasonal 1, 4) seasonal 2. Seasonal 1 used 10-15UTC in DJF, 9-16UTC in MAM and SON, and 8-17UTC in JJA. Seasonal 2 to used 10-15 in DJF, 9-17UTC in MAM and SON, and 7-19UTC in JJA (for the latter see Fig. R6).

The results for all of those experiments remain qualitatively the same. (For the 12UTC-only experiment only the probability was looked at, since the calculation of streak lengths like described in the manuscript does not make sense then). The conditional probability is almost undistinguishable, and the streak lengths (mean and 90th-percentile) increase slightly, which is logical, since the requirement of 3 h of lid per day is met more likely, if the days have more hours.

This analysis indicates that the results are robust towards changes in the time window. Hence, we want to keep the analysis the way it is in the paper. This is because we think adding a seasonal variation gives a false security, since local times and sun angle vary considerably across the domain as well. We clarify this in the manuscript as: "Our selection of 10–15 UTC does not aim to precisely capture the actual daytime hours at every location and day of the year. Instead, this time window is chosen because it represents a period during which the ERA5 BLH can be reasonably assumed to provide meaningful and useful information for our analysis. To ensure the robustness of our conclusions, we conducted the same analysis using alternative time windows, including a fixed time (12 UTC) and seasonally varying windows. The results remain consistent regardless of the exact choice of the daytime hours."

l. 173: "weighted by grid cell area". Even though you might use an area-weighted average for computation of the mean field values, I find this explanation quite confusing here. Figure 1 shows AD probability per hour and area (not per cell), and does not need further weighting for interpretation (I think).

I am not sure I understand what causes the confusion. The AD data is a binary time series for each grid cell. We calculate its temporal mean, which yields the occurrence probability per hour and cell. Since the cells hat higher latitude are smaller in area, it makes more sense to show occurrence probability per hour and area. To compute this, the occurrence frequency per hour and cell needs to be divided by the grid cell's area. This is what we meant by "weighted by grid cell area". We changed the wording to: "Due to Earth's curvature, ERA5 grid cells at high latitudes are smaller than those at lower latitudes. To account for that, the occurrence probability per grid cell is divided by the grid cells' area (Fig. 1)."

Lines 195-196: Do we know that a greater poleward extent equates to a longer duration? This seems counter to some of your other results. I would think the poleward extent vs. duration would highly depend on the synoptics that got the air to high latitudes in the first place (eg. a strong cyclone vs. a moderate anticyclone will have different advective velocities).

l. 145: Why 4 clusters? You say "as in Fix et al. (2024)", but please also give a brief reason in this paper why you choose this value.

"The variables are standardised to ensure equal weight and a number of 4 clusters is chosen for the clustering (as in Fix et al., 2024, where the ideal number of clusters was determined using the 'elbow-plot' of the total sum of squares)."

In Figure 2 long stripes mean the AD extends far northwards, wider stripes mean longer durations and darker colours mean wider longitudinal extent. As you can see, and as we mention 2 lines further, very different flavours of events exist - i.e. various combinations of latitudinal and longitudinal extent and duration happen. We do not claim anywhere that there is a connection between poleward extent and duration.

l. 198: "does show that it is rare that no AD air is present in the domain north of 370 N". It would be interesting to know how often this occurs over the analysed period.

In 2.7% of the hours during the analysed period, AD air is present in less than 1% of the columns in the domain. We add this information to the revised manuscript.

Lines 211-221; Figure 3: Aim to include panel labels here to clarify the discussion. In addition, I would consider either creating a second figure (or adding to figure 3) a composite of MSLP or 850 hPa heights here to capture the lower tropospheric flow. This matters both for the 24h after onset period (where the lower troposphere presumably has done much of the initial advection of the AD from Africa) and the latter period (which matters for your lower tropospheric trajectories). Consider including this and expanding/support the discussion accordingly.

850 hPa geopotential heights have been added to the respective figures and will be discussed in the improved manuscript. See Figs. R3 and R4.

Lines 223-229: Can these 'streaks' be reinforced by multiple synoptic scale events? It seems like a residence time of >7 days needs multiple events given the atmospheric variability in this region. As such, how does this impact the interpretation of your clusters?

Certainly, it can happen that AD come in several waves caused by multiple synoptic scale events. For the interpretation of the residence time of the AD here, that does not matter. We cluster the synoptic situation for the events based on the geopotential height maps 24 h before the maximum extent of the AD. This is comparable for all cases, independent of their length and whether they come in waves. We also display the synoptic situation at 24 h after the minimum, which shows the situation during the onset. If in a particular case the AD came in waves with very different synoptic patterns, we would miss this, yes. While analysing these rare cases might add minor details, it would not significantly alter the overall interpretation of the clusters or the conclusions of our study.

l. 229: Fig. 4: Reference this figure where you first describe streak lengths (previous paragraph). The paragraph from lines 223-229 is not about this Figure, since the Figure shows the seasonal results, while the previous paragraph discusses the results for the entire year, hence referencing it there would not make sense.

Line 233: Please change to '... below a week, but is closer to ...'

l. 233: "closer to a week in the Mediterranean in SON". Figure 4 shows less than one week (7 days) in the Mediterranean for SON. Please clarify.

The sentence is changed according to **reviewer2's** comment. Fig. 4f indeed shows 90th percentile values in the Mediterranean in SON are close to 7 days.

Line 243: I'm not sure I can see this north-east tilt from your figures. Also, given the spatial dimensions and rate of ascent here, how much of a horizontal extent of tilt should we expect to see?

The north-east tilt can be seen in Fig 5, because the colours become increasingly yellow towards the

north-east in panels a and b (colour scales will be improved).

The expected tilt really depends on the depth of the colder air mass the AD air encounters in the north, and is therefore difficult to estimate from the AD data alone.

l. 245: "The largest average distance between the highest and lowest AD-cells". Suggestion: The largest vertical depth . . .

We want to keep the wording, since 'vertical depth' can be misleading. 'Depth' might require continuos AD air in that layer, but the distance between highest and lowest does not.

**Line 247: Please remove 'in their centres'.**

We disagree with this comment. EMLs have been shown to inhibit thunderstorms at their centres while enhancing them near their edges; therefore, removing 'in their centres' would obscure the intended meaning.

Lines 256-267/methods: I found this definition of the lid challenging to feel 'comfortable' with. Why not go for a more rigorous definition like exceeding a threshold for vertical gradient in potential temperature? You have the data (given your later figures), and one would expect a larger magnitude vertical gradient in theta for a profile with an AD over the BL rather than one with a common free atmosphere overhead.

We agree that if AD air is present directly above the local BL, there should be a stably stratified layer. However, we do not investigate the strength of the AD-induced lids here, but only their occurrence. While for EMLs it can be expected that the theta gradient is strong, this is not necessarily the case for ADs, as they can be modified during the advection, so that their potential temperature can be closer to that of the local BL. Hence, we do not want to impose a threshold criterion on the theta gradient. Additionally, the definition of the ERA5 BLH itself implies the presence of a stable layer above, due to its use of the Bulk-Richardson Number.

Lines 256-258: Would it be better to show the fraction of the mean/90th percentile streak length that met the criteria of a lid instead? I feel like this may be a more natural extension/connection of the work.

Showing the comparison of AD streak lengths and lid streak lengths does not make sense like this. AD streak lengths are calculated as continuous phases of AD air being present in a specific column. Lid streak lengths, however, are calculated as the number of consecutive days that have a lid present for at least 3 h during daytime. Hence, streaks of lids are not necessarily a subset of those of AD streaks, since the AD streak could be interrupted between two consecutive days with a lid. This definition of lid streaks is useful, however, because for the lid's consequences it is important, whether it is present during several consecutive days during the daytime, independent of whether AD air covered the cell at all times.

Lines 278-282: Though heat waves matter here, please also consider adding discussion here on this being important for suppressing deep convection as well.

We added: "Sometimes the AD also forms a lid on top of the local BL, however, the persistence of these lids is typically short, so that they are not likely to cause considerable heat build-up and hence heat waves. Independent of its persistence, the presence of a lid can suppress or delay deep convection in its centre, and boost thunderstorm formation at its edges (as was shown for EMLs by e.g. Carlson and Ludlam, 1968; Keyser and Carlson, 1984; Lanicci and Warner, 1991b; Andrews et al., 2024), which

l. 265: "compared to a continental one, as it is often cooler and moister". What does this mean for the development and persistence of a lid? Please add/explain.

We added: "This likely has to do with the different properties of a marine BL compared to a continental one, as it is often cooler and moister. The greater the difference in thermodynamic properties between the AD air and the local boundary layer (BL), the less likely the AD is to intrude, making longer-lasting lids more probable."

**Line 315: I'm not sure I see this 'unchanged' characteristic, in particular for q.**

l. 315: Based on Fig. 8c, I struggle to see how q remains almost unchanged between 2 and 6 km. Or do you mean Theta\_E?

In Figure 8c) in the manuscript, a high number of cells have an average of near-zero change in the potential temperature, which can be seen in the dark colours at  $\Delta\theta = 0 \,\mathrm{K}$  which are darkest between 2 and about 6 km. Similarly, the darkest colours in Fig. 8d) can be found at  $\Delta q = 0 \,\mathrm{g\,kg^{-1}}$ , which is most visible between 2 and 4 km. We do not want to say that all cells in this altitude range retain their thermodynamic properties, but that there are many cells in this range, that do. This is rephrased as this section changed with the changes to Figure 9.

Section 3.5: Two points here to consider. First, I felt like theta-e (an excellent metric for an air mass) was presented but not really discussed. I think one key take-away here is that theta-e isn't really changing much (perhaps a slight weak bias toward a negative tendency), indicating that once the air mass is being advected, it's broad airmass characteristics are not really changing. In other words, changes to the moisture characteristics are being compensated by changes to the dry parcel characteristics, resulting in a nearly-conserved air mass overall property. The second point here (noted in the general comments above) – it would be really helpful to see how your trajectories here relate to the 500 hPa composites. There are a variety of ways to approach this, but I think it would really help connect the different analysis approaches of the manuscript well.

Thank you very much for this comment. Indeed, we will discuss  $\theta_E$  in more detail: "Throughout the column, the average changes are almost zero (with a slight tendency towards negative changes), which indicates that overall the air mass properties are preserved well. When looking at the changes for the trajectory clusters, often the yellow and green air streams present with large changes in  $\theta_E$  (see Supplement), which is logical, since their changes are dominated by adiabatic processes."

We have added the trajectory cluster means for the cluster centroid closest days to what was Fig 3. in the manuscript (Fig. R3) and mention how they connect to the respective synoptic maps.

**Specific Comments on Figures**

Colour bars have been adjusted on all relevant figures to improve readability.

**Figure1**

Caption: You mention cells, but your colourbar legend shows area im  $km^2$ . Please clarify. "Probability of an AD being present in the respective cell, weighted by the cell's area" is what it says in the caption, which explains why the unit has  $km^{-2}$ . We do not think further clarification is needed.

**Figure 3**

It would be very interesting to see a seasonality analysis of these clusters. See earlier answer.

**1 Figures**

Figure R1: Explained variance ratio for the principal components. Based on standardised  $Z_{500}$  anomalies as input.

Figure R2: Elbow plot of the total sum of squares (inertia). Based on clustering of 7 PCs based on standardised anomalies of Z500.

Figure R3:  $Z_{500}$  in m (green contours),  $Z_{850}$  in m (blue contours) and AD-columns (grey shading) for the centroid closest observation for the four clusters, respectively (columns). The top row (a-d) shows the situation at 24 h after the onset, and the second row (e-h) at 24 h before the AD maximum. The AD-columns are shown as shading for context. Clusters are based on standardised  $Z_{500}$  anomalies only at 24 h before the maximum. The bottom row (i-l) shows the four trajectory clusters based on the trajectories started on the day 5 days prior to the AD maximum, colouring of the trajectory clusters is according to the classification algorithm. The number of members in each cluster is given in the column title.

Figure R4: As Fig. R3 but for cluster averages. Panels i)-l) show the seasonal breakdown of each cluster in percent.

Figure R5: Conditional probability of an anomaly of dust aerosol optical depth (550nm) being greater than 1 (duaod550' > 1), given the presence of AD air (a), conditional probability of duaod500' > 1, given the absence of AD air (b), and the logarithm of their odds ratio (c).

Figure R6: As Figure 6 in manuscript, but for seasonally varying daytime hours. Lid properties. (a) Probability of a lid being present, given an AD is present. (b) Mean streak length of the lid in days. (c) 90th percentile of streak length of the lid in days.

---

## Referee Report (RR1)

**Review of 1st revision to WCD-2025-3552:**
**Properties and Characteristics of Atmospheric Deserts over Europe - A First Statistical Analysis**

**Summary**
I reviewed an earlier version of this manuscript. The authors have made comprehensive improvements that significantly improved the quality of their manuscript. Especially the enhancements to the cluster analysis have improved the robustness of their results, and their visualisation in new Figure 3 with the inclusion of seasonal statistics is a great addition to the work.
Improved language, clarifications, improved figures and the extended manuscript title successfully address all my previous major concerns. I only have a two minor comments to the revised version. I  recommend the manuscript to be accepted for publication once these points are resolved:

- The extended discussion of impacts from AD events and agreement with other studies' results on Saharan warm-air intrusions and dust transport is very useful for understanding the wider relevance of the authors' research. However, if mentioning atmospheric rivers (AR) it should be noted  that ARs also often coincide with Saharan dust transport to Europe (e.g., Francis et al. 2022).
  I think it would be valuable to explicitly state that this is not a contradiction to the author's results, due to the different definition of AR and AD (ARs can be ADs).

  Francis, D., Fonseca, R., Nelli, N., Bozkurt, D., Picard, G., & Guan, B. (2022). Atmospheric rivers drive exceptional Saharan dust transport towards Europe. Atmospheric Research, 266, 105959.

- In your reply to my previous comment on Figure 4 you state that "Fig. 4f indeed shows 90th percentile values in the Mediterranean in SON are close to 7 days". However, this still does not agree with what I read from Figure 4:
  - Fig. 4**f** shows  90th percentile streak length  for **JJA**, where the Mediterranean indeed shows streak length close to 7 days. I cannot see from this figure whether this corroborates your statement in line 277 of "more than nine days" since the colour scale is maxed out at 7+ days.
  - Fig.4**g** shows  90th percentile streak length for **SON** around  3 days in the Mediterranean.
  Please clarify and refer to correct season or (sub)figure in your manuscript.

---

## Author Response (AR3)

**Author's Response Final**

Dear Christian Grams,
Dear reviewers,

thank you very much for all the feedback and the acceptance.
Please find below the comments to the second round of reviews again as in the previous upload.

Thank you very much,
Fiona Fix-Hewitt and Co-Authors

We addressed all issues:
1. We double checked the data in Fig. 4 and rephrased the sentence in question:
> "The 90th percentile of streak length varies strongly across the seasons (Fig. 4, bottom row). In most seasons, the 90th percentile of the streak length is well below a week, but reaches a week in the south-western Mediterranean in SON. During JJA, however, 90th percentile streak lengths of more than a week for most of the Mediterranean Sea (even exceeding nine days in some cells, not shown) are reached in the Mediterranean, maxima are even up to four weeks." (l 275 ff.)

2. We inserted the sentence:
> "Also atmospheric rivers can carry dust, as they have been described to pick up dust in the Saharan boundary layer while travelling from the Atlantic towards the continent (Dezfuli et al., 2021; Francis et al.,2022). Hence, it is possible for an atmospheric river to also be an AD, which is not a contradiction but simply reflects the different definitions of the two terms." (l. 446ff)

3. We mention the dusty-cirrus topic:
> "Dusty cirrus clouds can occur when Saharan dust ascends into the upper troposphere (as it can with warm conveyor belts, or the ascending air stream of an AD; Fromm et al., 2016; Seifert et al., 2023; Hermes et al., 2024), forming dust-infused baroclinic storms with characteristic cirrus decks over Europe. These dust events can modify shortwave and longwave radiation fluxes and cloudiness through direct, indirect, and semi-direct dust effects (Helmert et al., 2007) and hence additionally change the vertical temperature profile. Forecasting the cloud and radiative effects of dust events poses a challenge to numerical weather prediction models, which often do not include dust prognostically, and also lack proper parametrisation of dust-cloud-radiation interactions (Hermes et al., 2024; Seifert et al., 2023). Including prognostic dust and a proper parametrisation to describe dusty cirrus effects has been shown to improve forecasts of clouds and radiative fluxes considerably (Hermes et al., 2024; Seifert et al., 2023). This highlights the relevance of AD-linked dust events, especially for cloud and radiation forecasts, which are especially important to the renewable energy sector. All the above mentioned consequences of ADs remain to be investigated in detail." (l. 433 ff.)